# Human kidney is a target for novel severe acute respiratory syndrome coronavirus 2 infection

Bo Diao[1,2,3,9], Chenhui Wang[1,9], Rongshuai Wang[4,9], Zeqing Feng[1], Ji Zhang[1], Han Yang[1], Yingjun Tan[2], Huiming Wang[5], Changsong Wang[6], Liang Liu[7], Ying Liu[2], Yueping Liu[2], Gang Wang[2], Zilin Yuan[2], Xiaotao Hou[8], Liang Ren[4,9✉], Yuzhang Wu[1,9✉] & Yongwen Chen[1,9✉]

It is unclear whether severe acute respiratory syndrome coronavirus 2 (SARS-CoV-2) can directly infect human kidney, thus leading to acute kidney injury (AKI). Here, we perform a retrospective analysis of clinical parameters from 85 patients with laboratory-confirmed coronavirus disease 2019 (COVID-19); moreover, kidney histopathology from six additional COVID-19 patients with post-mortem examinations was performed. We find that 27% (23/85) of patients exhibited AKI. The elderly patients and cases with comorbidities (hypertension and heart failure) are more prone to develop AKI. Haematoxylin & eosin staining shows that the kidneys from COVID-19 autopsies have moderate to severe tubular damage. In situ hybridization assays illustrate that viral RNA accumulates in tubules. Immunohistochemistry shows nucleocapsid and spike protein deposits in the tubules, and immunofluorescence double staining shows that both antigens are restricted to the angiotensin converting enzyme-II-positive tubules. SARS-CoV-2 infection triggers the expression of hypoxic damage-associated molecules, including DP2 and prostaglandin D synthase in infected tubules. Moreover, it enhances CD68+ macrophages infiltration into the tubulointerstitium, and complement C5b-9 deposition on tubules is also observed. These results suggest that SARS-CoV-2 directly infects human kidney to mediate tubular pathogenesis and AKI.

[1] Institute of Immunology, PLA, Third Military Medical University, Chongqing, P. R. China. [2] Department of Medical Laboratory Center, General Hospital of Central Theater Command, Wuhan, Hubei Province, P. R. China. [3] Hubei Key Laboratory of Central Nervous System Tumor and Intervention, Wuhan, Hubei Province, P. R. China. [4] Department of Forensic Medicine, Tongji Medical College, Huazhong University of Science and Technology, Wuhan, Hubei Province, P. R. China. [5] Department of Nephrology, Renmin Hospital of Wuhan University, Wuhan, Hubei Province, P. R. China. [6] Department of Pathology, 989th Hospital of PLA, Luoyang, Henan Province, P. R. China. [7] Hubei Chongxin Judicial Expertise Center, Wuhan, Hubei Province, P. R. China. [8] Guangzhou KingMed Center for Clinical Laboratory Co., Ltd., Guangzhou, Guangdong Province, P. R. China. [9] These authors contributed equally: Bo Diao, Chenhui Wang, Rongshuai Wang, Liang Ren, Yuzhang Wu, Yongwen Chen. ✉email: 36918280@qq.com; wuyuzhang@tmmu.edu.cn; yongwench@163.com

In December 2019, a cluster of pneumonia cases caused by a novel severe acute respiratory syndrome coronavirus 2 (SARS-CoV-2) were reported in Wuhan, Hubei Province, China[1–3]. This disease, now designated as coronavirus disease 2019 (COVID-19) by the WHO, rapidly spread to other cities of China and then around the world[4–6]. According to the daily reports of the WHO, the epidemic of SARS-CoV-2 has so far caused 108.2 million cumulative COVID-19 cases and 2.3 million deaths globally by February 2021[7]. The typical symptoms of COVID-19, including fever, cough, shortness of breath, and pneumonia, are very common in mild cases[8,9], whereas acute respiratory distress syndrome (ARDS), acute cardiac injury, and lymphocytopenia are also prevalent, especially in aged and critically ill cases[8–10], implying that SARS-CoV-2 affects various organs.

Some studies have suggested that SARS-CoV-2, SARS-CoV (severe acute respiratory syndrome-coronavirus) and MERS-CoV (Middle Eastern respiratory syndrome-coronavirus) share a common ancestor resembling the bat coronavirus HKU9-1[11,12]. Recently, Shi et al., reported that SARS-CoV-2 interacts with human angiotensin-converting enzyme-II (ACE2) molecules via its Spike (S) protein[13]. The crystal structure of the C-terminal domain of SARS-CoV-2 (SARS-CoV-2-CTD) spike protein in complex with human ACE2 (hACE2) reveals a hACE2-binding mode similar overall to that observed for SARS-CoV[14]. Actually, ACE2 is widely expressed in the heart, vessels, gut, lung (particularly in type 2 pneumocytes and macrophages), kidney, testis, and brain[15,16]. Recently, Cheng et al., have reported that some COVID-19 patients had an elevated level of proteinuria, suggesting SARS-CoV-2 infection might result in acute kidney injury (AKI)[17]. SARS-CoV-2 can also directly infect human kidney organoids via the ACE2 receptor[18]. However, no evidence has demonstrated that SARS-CoV-2 can directly infect human kidney leading to AKI.

In this work, we retrospectively analyse the clinical data on renal function from 85 cases of COVID-19 who were admitted into the General Hospital of Central Theatre Command in Wuhan, China, from 17 January 2020 to 13 February 2020. To elucidate the mechanism of renal injury caused by SARS-CoV-2 infection, we also use haematoxylin & eosin (H&E) staining, in situ hybridization (ISH), immunohistochemistry, and immunofluorescence to visually assess the histopathology and viral presence in kidney tissues from six additional COVID-19 autopsies from Jinyintan Hospital, another designated hospital in Wuhan.

## Results

**SARS-CoV-2 induces AKI in COVID-19 patients**. We first retrospectively analyzed 85 cases of COVID-19 patients who had clinical data of kidney function, including estimated glomerular filtration rate (eGFR), serum urea nitrogen (Urea), creatinine, and other clinical parameters from the General Hospital of Central Theatre Command in Wuhan, Hubei Province, China. We found that 27% (23/85) of COVID-19 patients exhibited AKI based on eGFR, a critical parameter indicating renal function. High mortality and a more severe clinical type overlapped with AKI, although no significant difference was detected. Patients who were elderly (≥60 years, 65% vs 24%, $p < 0.001$) or had comorbidities (70% vs 11%, $p < 0.001$) such as hypertension (39% vs 13%, $p = 0.0007$) and coronary heart disease (21.74% vs 4.84%, $p = 0.018$) were more likely to develop AKI (Table 1). The levels of serum creatinine and Urea were consistent with eGFR, which changed more obviously in the AKI group (Table 1). High urine glucose and electrolyte disorder were also observed in AKI patients, suggesting tubular injury rather than glomerular dysfunction (Table 1, Supplementary Table 1). These 85 patients

### Table 1 Clinical characteristics of COVID-19 patients in this study.

| Group | AKI | Non-AKI | p value | Discovery |
|---|---|---|---|---|
| | (N = 23) | (N = 62) | | |
| *Age* | | | | |
| ≥60 yr | 15 (65.22) | 15 (24.19) | <0.001 | * |
| <60 yr | 8 (34.78) | 47 (75.81) | | |
| *Gender* | | | | |
| Male | 15 (65.22) | 33 (53.23) | 0.322 | |
| Female | 8 (34.78) | 29 (46.77) | | |
| *Admission to AKI, day* | 1.00 (0.00–1.00) | | | |
| *Clinical outcome* | | | | |
| Discharge | 20 (86.96) | 60 (96.77) | 0.087 | |
| Perish | 3 (13.04) | 2 (3.23) | | |
| *Clinical classfication* | | | | |
| Mild/moderate | 12 (52.17) | 45 (72.58) | 0.071 | |
| Severe | 5 (21.74) | 12 (19.35) | | |
| Critical | 6 (26.09) | 5 (8.06) | | |
| *Coexisting disorder* | | | | |
| Any | 16 (69.57) | 7 (11.29) | <0.001 | * |
| Hypertension | 9 (39.13) | 8 (12.90) | 0.007 | * |
| Coronary heart disease | 5 (21.74) | 3 (4.84) | 0.018 | * |
| other heart disease | 4 (17.39) | 4 (6.45) | 0.125 | |
| Diabetes | 3 (13.04) | 4 (6.45) | 0.326 | |
| COPD | 2 (8.70) | 0 (0.00) | 0.019 | * |
| Chronic renal disorder | 2 (8.70) | 3 (4.80) | 0.502 | |
| Cerebrovascular disease | 1 (4.35) | 2 (3.23) | 0.803 | |
| Cancer | 1 (4.35) | 2 (3.23) | 0.803 | |
| Hepatitis B infection | 0 (0.00) | 3 (4.84) | 0.283 | |
| *Anti-hypertensive medication* | | | | |
| Any | 9 (39.13) | 7 (11.29) | 0.004 | * |
| α Recepter Blocker | 1 (4.35) | 2 (3.23) | 0.803 | |
| ACEI | 2 (8.70) | 0 (0.00) | 0.019 | * |
| ACEI in Hypertension | 2 (22.22) | 0 (0.00) | 0.156 | |
| ARB | 4 (17.39) | 4 (6.45) | 0.125 | |
| ARB in Hypertension | 4 (44.44) | 4 (50.00) | 0.82 | |
| β Recepter Blocker | 3 (13.04) | 4 (6.45) | 0.326 | |
| CCB | 8 (34.78) | 5 (8.06) | 0.002 | * |
| Diuretic | 6 (26.09) | 5 (8.06) | 0.028 | * |
| *Laboratory findings* | | | | |
| Creatinine | 84 (67–104.5) | 60.5 (51–72) | 0.004 | * |
| eGFR | 78.0 (64.3–89.6) | 121.5 (102.3–139.5) | <0.001 | * |
| Urea | 6.1 (4.5–10.7) | 3.7 (3.3–4.7) | <0.001 | * |
| Uroketone body (KET) | 6/17 | 12/51 | 0.341 | |
| Urine blood (BLD) | 4/17 | 16/51 | 0.539 | |
| Urine protein(PRO) | 7/17 | 13/51 | 0.219 | |
| Urine nitrite (NIT) | 1/17 | 1/51 | 0.407 | |
| Urinary leukocyte | 5/17 | 5/51 | 0.048 | * |
| Urine glucose(GLU) | 7/17 | 6/51 | 0.007 | * |

*p < 0.05.
Data are n (%), median(IQR) or n/N. p values (two side) are from χ2 or unpaired t test.
*ACEI* angiotensin-converting enzyme inhibitors, *AKI* acute kidney injury, *ARB* angiotensin receptor blocker, *CCB* calcium channel blocker, *COPD* chronic obstructive pulmonary disease, *yr* years.
Source data are provided as a Source data file.

were further categorized into three groups (mild/moderate, severe, and critical) by their symptoms and clinical signs, and the patients in the critical group had significantly lower levels of eGFR but dramatically higher serum concentrations of creatinine and Urea than those in the mild/moderate groups (Supplementary Fig. 1), suggesting AKI is relatively common in critically ill COVID-19 patients.

Angiotensin-converting enzyme inhibitors/angiotensin II receptor blockers (ACEI/ARB) are widely used as antihypertensive medication. We explored whether these medications could have side functions to aggravate the development of AKI. Although only some of the patients used ACEI or ARB, making this hard to evaluate, it seems that ACEI/ARB treatment does not lead to AKI, especially in hypertensive patients (Table 1). This result is consistent with a recent study reporting that ACEI and ARB did not affect the mortality of patients with hypertension hospitalized with COVID-19[19].

**Kidney tissues from COVID-19 autopsies present evidence of severe tubular injury.** We next examined whether SARS-CoV-2 might promote renal damage, and the kidneys from six COVID-19 subjects with post-mortem examinations in Jinyintan Hospital were collected. The post-mortem cause of death was due to ARDS. These cases were also given empirical antimicrobial treatment and antiviral therapy during the course of hospitalization; however, none of them received extracorporeal membrane oxygenation treatment. Other clinical characterizations of these six cases are shown in Table 2.

The histopathological examination was performed via H&E staining. These six renal specimens showed varying degrees of tubular necrosis, luminal brush border sloughing, vacuole degeneration, and leukocyte infiltration. Severe glomerular injury was absent in all cases, although benign hypertensive glomerulosclerosis was noted in four hypertensive patients (case #1, #2, #4, and #5). No obvious leukocyte infiltration was identified in the tubulointerstitium, but tubular damage or glomerular injury was observed in kidney sections from trauma victims (Fig. 1). The histopathology of kidney biopsies from hepatitis B virus-associated membranous nephropathy (HBV-MN) patients, on the other hand, showed that the glomerulus was affected by segmental necrosis or segmental sclerosis, whereas the renal tubules were apparently normal, and no obvious atrophy or necrosis was observed (Fig. 1).

According to the aforementioned histopathology results, patients infected by SARS-CoV-2 presented kidney defects that appear different from nephritis and nephrotic syndrome, in which the pathological signs are mainly shown in the glomerulus, rarely accompanied by tubular injury. The signs are also different from toxic tubular injury, which manifests with exfoliation and necrosis of tubular cells and interstitial oedema but not leukocyte infiltration. Moreover, ischaemic acute tubular necrosis (ATN) has damage in all renal tubules, but basement membrane damage is also common[20], which was rarely detected in these COVID-19 autopsies.

To further understand the influence of age and coexisting disorders on renal injury, we compared two pairs of autopsies (#2 vs. #6 and #1 vs. #4) specifically. The kidney sections from case #2, an 86-year-old female patient with hypertension and coronary disease for more than 10 years, manifested obvious glomerulosclerosis, protein accumulation within the tubules, severe interstitial fibrosis, along with acute pathological signs such as tubular epithelial cells exfoliation, necrosis, vacuolation, and strong leukocyte interstitial infiltration. However, sections from case #6, a 67-year-old female patient without coexisting disorders, showed scarcely any glomerular changes and milder luminal brush border sloughing and tubular necrosis. Correspondingly, compare case #1, a 70-year-old male patient with hypertension for more than 10 years, with #4, a 51-year-old male patient with hypertension for over 10 years. Both have glomerulosclerosis, luminal brush border sloughing, epithelial cells exfoliation, necrosis, and leukocyte interstitial infiltration, although the signs of case #4 are slightly milder than those in case #1 (Fig. 1, Table 2). All of these results hint that age and coexisting disorders could both exacerbate renal injury. More specifically, coexisting disorders, such as hypertension, are the more serious risk factor, and the glomerular changes are more likely attributable to the underlying diseases.

**SARS-CoV-2 directly infects human kidney tubules.** To confirm that SARS-CoV-2 directly infects human kidney thus lead to AKI, RNA ISH was performed using RNAScope probes directed against SARS-CoV-2 *S* gene, targeting 21631–23303 of NC_045512.2 base pairs (RNAScope® Probe-V-nCoV2019-S, #848561, ACD, Newark, CA) and RNAScope® Probe-V-nCoV2019-S-sense (#845701, ACD). The results showed that all of kidney tissues from COVID-19 autopsies manifested positive signals for Probe-V-nCoV2019-S, and the positive expression was found on the tubules and some infiltrating cells, demonstrating SARS-CoV-2 directly infects human kidney tubules. Interestingly,

**Table 2 The pathologic findings of kidney tissues from COVID-19 patients.**

| Case number | #1 | #2 | #3 | #4 | #5 | #6 |
|---|---|---|---|---|---|---|
| Cause of death | ARDS | ARDS | ARDS | ARDS | ARDS | ARDS |
| *Coexisting disorder* | | | | | | |
| Hypertension | >10 yr | >10 yr | – | >10 yr | >5 yr | – |
| Coronary disease | – | >10 yr | – | – | – | – |
| Gout | – | – | – | – | Yes | – |
| *H&E staining* | | | | | | |
| Benign glomerulosclerosis | + | + | – | + | + | – |
| Brush border exfoliation | +++ | +++ | + | + | ++ | ++++ |
| Tubular necrosis | +++ | +++ | ++ | ++ | ++ | ++++ |
| Interstitial infiltration | +++ | +++ | ++ | ++ | ++ | – |
| Interstitial fibrosis | – | ++ | – | – | ++ | – |
| Autolysis | ++ | + | + | + | + | + |
| Vascular congestion | + | ++ | – | ++ | +++ | – |
| *IHC/IF* | | | | | | |
| Viral NP antigen | +++ | ++++ | +++ | +++ | +++ | +++ |
| C5b-9 on tubules | ++ | ++++ | ++ | + | ++ | + |
| DP2 | No test | ++ | +++ | ++ | No test | No test |
| PDS | No test | +++ | +++ | ++ | No test | No test |
| *SARS-CoV-2 viral RNA* | | | | | | |
| Probe-V-nCoV2019-S | No test | +++ | No test | No test | +++ | ++ |
| Probe-V-nCoV2019-S-sense | No test | +++ | No test | No test | ++ | ++ |

−: negative; +: <10% tubules positive; ++: 10%–30% tubules positive; +++: 30%–50% tubules positive; ++++: >50% tubules positive.
*ARDS* acute respiratory distress syndrome, *H&E* haematoxylin & eosin, *IHC/IF* immunohistochemistry/immunofluorescence, *NP* nucleocapsid protein, *PDS* prostaglandin D synthase, *TEM* transmission electronic microscopy, *yr* years.
Vrial NP antigen was detected by anti-SARS NP antibodies (clone ID: 019, rabbit IgG; Sino Biological).

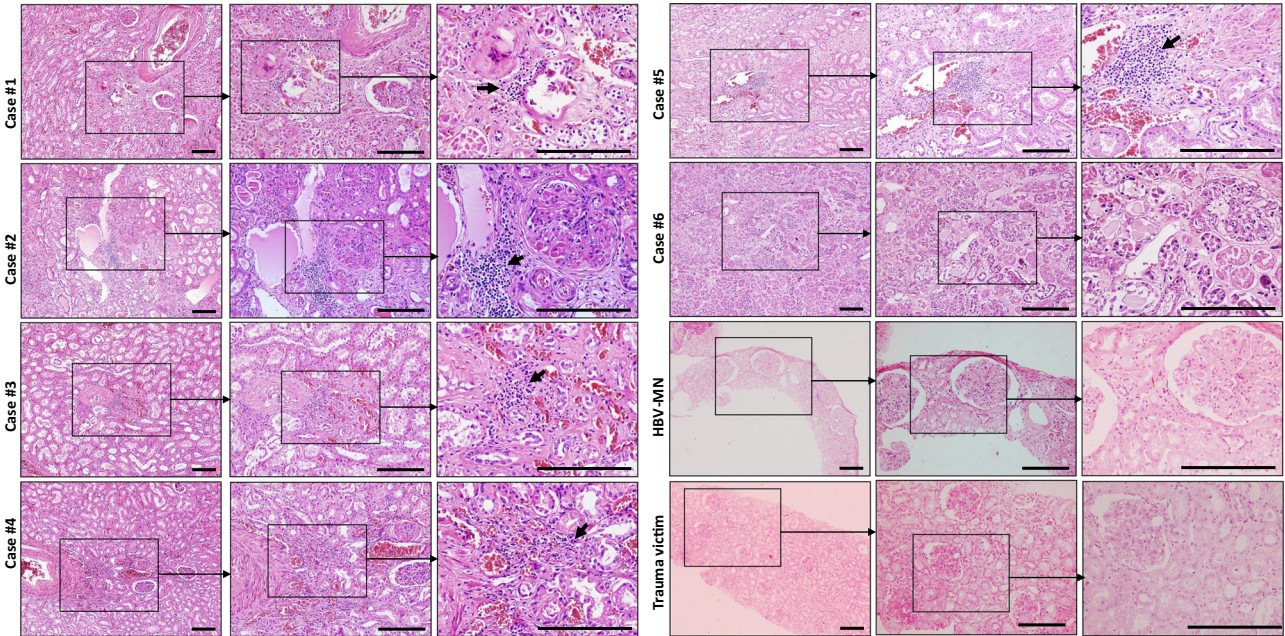

**Fig. 1 The histopathology of kidney tissues from COVID-19 post-mortem.** Sections from six cases of COVID-19 post-mortem, one representative hepatitis B virus-associated membranous nephropathy (HBV-MN) patient and one representative trauma victim were collected to stain by haematoxylin & eosin (H&E). Arrow indicates infiltrated leukocytes. Scale bars = 200 μm. Data represent one of at least three technical replication each.

some tubules are also positive for Probe-V-nCoV2019-S-sense, illustrating that SARS-CoV-2 has the capacity to replicate in vivo. Conversely, no signals for Probe-V-nCoV2019-S nor Probe-V-nCoV2019-S-sense were observed in the kidney tissues from HBV-MN patients (Supplementary Fig. 2). Here a negative control probe to the bacterial gene diaminopimelate B (DapB) and a positive control probe to the housekeeping gene peptidylprolyl isomerase B (PPIB) for RNA integrity were also included (Fig. 2a, Table 2).

The expression of SARS-CoV-2 nucleocapsid protein (NP) and spike (S) antigens in kidney tissues were further assessed by immunohistochemistry, showing SARS-CoV-2 NP and S antigens could be seen in kidney tissues from all six samples, with their expression restricted to the kidney tubules, while NP and S antigens were absent in the glomerulus (Fig. 2b). Conversely, both NP- and S- antigens were absent in kidney tissues from trauma victims and HBV-MN patients (Supplementary Fig. 3). Here, sections treated with rabbit or mouse isotype control antibodies were used as isotype controls (Fig. 2b), and the expression of SARS-CoV-2 NP antigen in lung tissues from COVID-19 autopsy (case #2) was used as a positive control (Supplementary Fig. 4).

Previous studies have shown that the SARS-CoV-2 receptor ACE2 is expressed on human kidney tubules[21]. Similar to that work, immunohistochemistry showed that kidney tissues from COVID-19 post-mortem have ACE2 expression, and the expression is major in brush border of proximal tubular cells (Fig. 3a). Immunofluorescent (IF) double staining illustrated that both SARS-CoV-2 viral NP and S antigens are co-localized with the ACE2 protein in the kidney tubules (Fig. 3b). The sections from HBV-MN and trauma victims' biopsies are also positive for ACE2, however, these sections are rarely stained by anti-NP or anti-S antibodies, especially in the kidney tubules (Fig. 3b). Collectively, these data suggest that SARS-CoV-2 directly infects human kidney tubules through the ACE2 receptor.

**SARS-CoV-2 directly induces tubular damage.** We next explored the pathological pathways involved in the kidney injury. The expression of hypoxic damage-associated molecules,

including DP2 and prostaglandin D synthase (PDS), in the kidney tissues was detected by immunofluorescence staining. Both of these molecules showed high expression in kidney tubules from COVID-19 autopsies, and they were co-expressed with SARS-CoV-2 specific NP and S antigens (Fig. 4a, b). Conversely, sections from HBV-MN did not present any DP2 or PDS antigens (Fig. 4c), suggesting SARS-CoV-2 virus can directly initiate hypoxic damage in infected kidney tubules.

The leukocyte-derived proinflammatory cytokines, such as interleukin (IL)-6, IL-1β, and tumor necrosis factor-alpha (TNF-α) can induce cell apoptosis and pyroptosis, thus accelerating tubular damage and interstitial fibrosis[22]. Our data mentioned above also showed that kidney tissues from COVID-19 autopsies have strong or moderate leukocyte infiltration (Fig. 1). We next examined the identity of these infiltrated cells, and immunohistochemistry showed strong CD68+ macrophages were seen in the tubulointerstitium of these six autopsies, while moderate numbers of CD8+ T cells were observed in the tubulointerstitium of two cases, whereas low number of CD56+ natural killer (NK) cells were found in the examined COVID-19 post-mortem tissues (Fig. 5a). The distribution of these cells in sections from HBV-MN and trauma victims was also investigated, and the results showed that sections from HBV-MN cases have CD68+ macrophages, CD8+ T cells and CD56+ NK cells distribution, even the counts are very low, whereas the distribution of these cells in sections from trauma victims are not obvious (Fig. 5a). Collectively, these results suggest that SARS-CoV-2 infection majorly recruits CD68+ macrophages into the tubulointerstitium.

As the deposition of the complement membrane-attack complex (MAC, also named C5b-9) on tubules or glomeruli may also cause renal damage[23], we further analyzed C5b-9 status in kidney tissues from these post-mortem cases. Interestingly, strong C5b-9 depositions on the tubules were observed in all six cases, and two cases manifested with low levels of C5b-9 deposition on the glomeruli and capillaries (Fig. 5b, Table 2). However, C5b-9 deposition was absent in sections from trauma victims, whereas sections from HBV-MN manifested slight C5b-9 deposition in tubules (Fig. 5b), suggesting both SARS-CoV-2 can

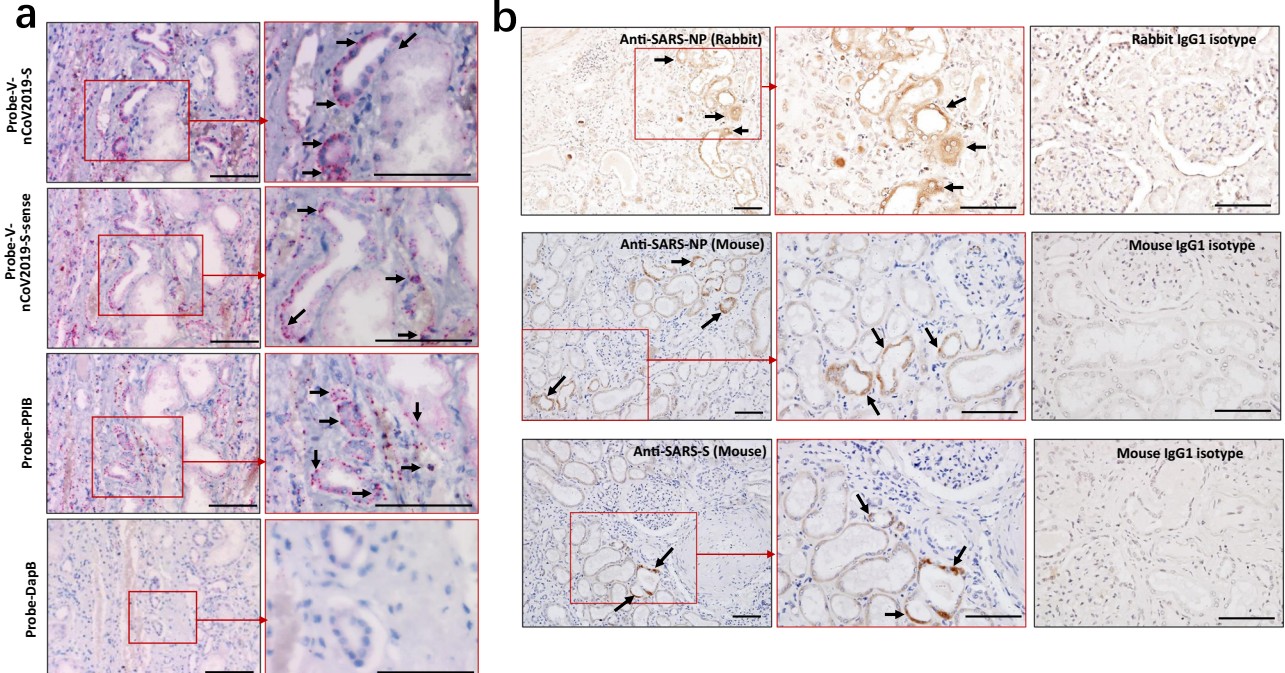

**Fig. 2 Detection of SARS-CoV-2 viral RNA and protein antigens in kidney tissues. a** RNA in situ hybridization (ISH) in serial sections of kidney tissues from COVID-19 post-mortem (case #2) showed reactivity for the presence of SARS-CoV-2 viral RNA and the positive control probe to the housekeeping gene peptidylprolyl isomerase B (PPIB). Scale bars = 100 μm. **b** The kidney tissues from COVID-19 post-mortem (case #2) were incubated with primary anti-SARS-NP (nucleocapsid protein) antibodies (clone ID: 019, rabbit IgG; Sino Biological), anti-SARS-NP antibodies (ab273434, mouse monoclonal 6H3, Abcam), anti-SARS-S (spike) antibodies (ab273433, mouse monoclonal 1A9, Abcam), or only incubated with mouse or rabbit IgG1 control antibodies, and the expression of indicated viral antigens were further analyzed by immunohistochemistry. Arrow indicates positive tubules. Scale bars = 100 μm. Data represent one of at least three technical replication each.

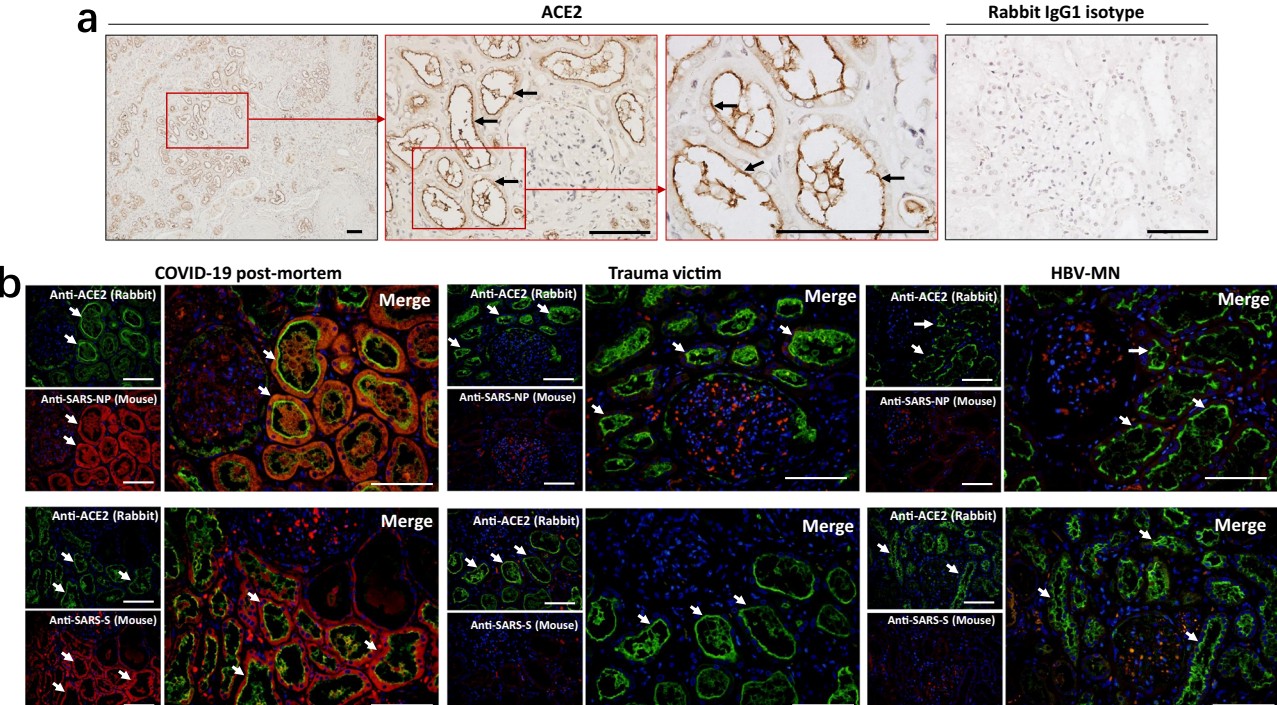

**Fig. 3 SARS-CoV-2 protein antigens co-express with ACE2 in kidney tubules. a** Representative expression of angiotensin-converting enzyme-II (ACE2) in kidney tissues from COVID-19 post-mortem (case #2) was detected by immunohistochemistry. Scale bars = 100 μm. **b** The co-expression of ACE2 and SARS-CoV-2 NP (nucleocapsid protein) or S (spike) antigens in kidney sections from COVID-19 post-mortem (case #2), hepatitis B virus-associated membranous nephropathy (HBV-MN) and trauma victims. Arrow indicates positive tubules. Scale bars = 100 μm. Data represent one of at least three technical replication each.

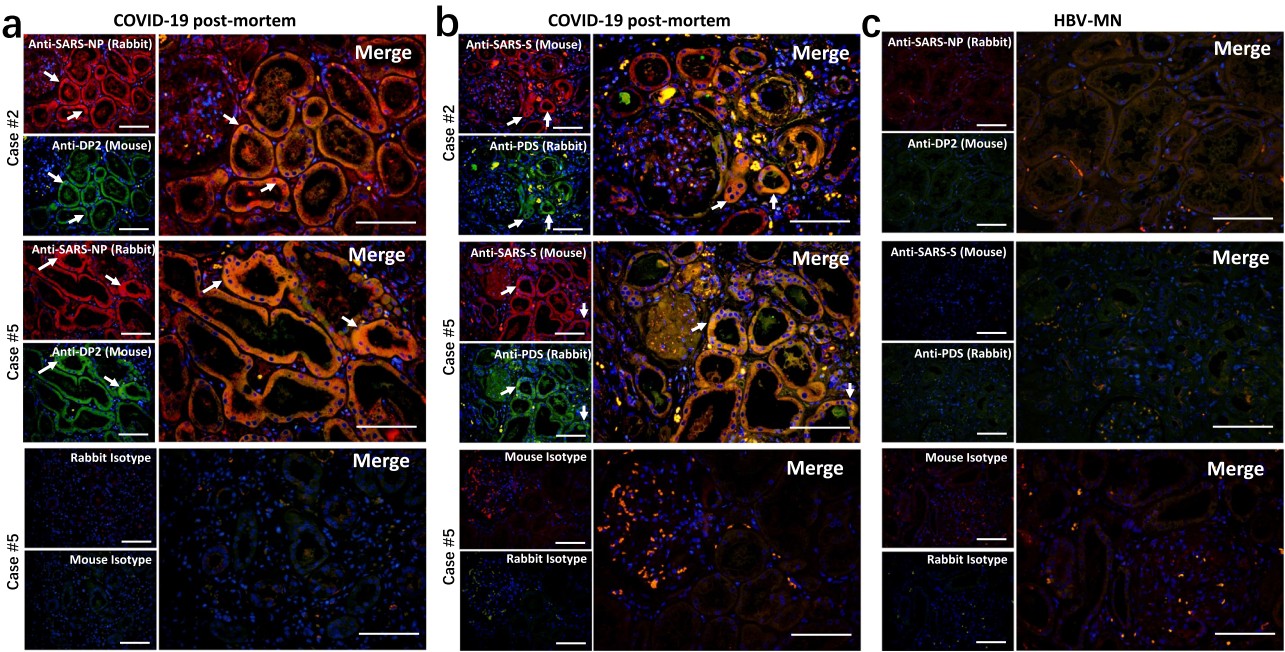

**Fig. 4 SARS-CoV-2 protein antigens co-express with hypoxic damage-associated molecules in kidney.** Immunofluorescent double staining was used to detect the expression of **a** SARS-CoV-2 NP (nucleocapsid protein) and DP2 in kidney tissues from COVID-19 post-mortem (case #2 and #5); **b** SARS-CoV-2 S (spike) and prostaglandin D synthase (PDS) in kidney tissues from COVID-19 post-mortem (case #2 and #5); **c** SARS-CoV-2 NP and DP2 or S and PDS in kidney tissues from hepatitis B virus-associated membranous nephropathy (HBV-MN). Here, sections only incubated with rabbit and mouse secondary antibodies were used as isotype controls. Arrow indicates positive cells. Scale bars = 100 μm. Data represent one of at least three technical replication each.

induce kidney tubular lesions through triggering C5b-9 deposition.

## Discussion

The available data revealed that proteinuria and haematuria are common features of COVID-19 patients on hospital admission, and computerised tomography scans of the kidneys from COVID-19 cases showed reduced density, suggestive of inflammation and oedema[24]. A very recent study by Yao et al. confirmed that SARS-CoV-2 damages kidney, and SARS-CoV-2 infection induces AKI based on a small subset of COVID-19 patients[25]. In this study, we found that 27% of COVID-9 patients had abnormal AKI, and patients who are aged or have coexisting disorders such as hypertension and heart failure more commonly developed AKI (Table 1), suggesting that AKI is relatively common following SARS-CoV-2 infection. This phenomenon is different from SARS, in which AKI was rare, despite being one of the top risk factors for mortality[26]. Yan et al., reported that 63% (32/51) of COVID-19 patients had an elevated level of proteinuria[13]. Similar to this work, we showed here that 29% (20/68) cases manifested higher levels of eGFR (Table 1 and Supplementary Fig. 1). Additionally, the kidney tissues from COVID-19 autopsies manifested severe tubular damage and strong leukocyte infiltration (Fig. 1). Collectively, these results illustrate that SARS-CoV-2 mediated AKI may be one of the major causes of multi-organ failure and eventual death of COVID-19 patients.

To further analyse how SARS-CoV-2 affects renal function, we focused on analysing renal tissue morphology from COVID-19 autopsies. H&E staining showed that acute renal tubular damage and leukocyte infiltration was observed in all six cases, while the glomeruli were intact, except for slight glomerulosclerosis was found in some cases who had coexisting disorders (Fig. 1), suggesting that other conditions such as hypertension and diabetic nephropathy may have been involved in the pathogenesis. RNA

ISH illustrated that SARS-CoV-2 viral RNA, immunohistochemistry, and IF showed that both viral NP and S protein antigens are all found in the kidney tubules from COVID-19 autopsies (Figs. 2, 3), suggesting that SARS-CoV-2 directly infects human kidney. This phenomenon is consistent with MERS-CoV, which can infect human kidneys[27,28]. The kidneys from SARS-autopsies have been observed to be focally haemorrhagic and have various degrees of acute tubular necrosis, however, no glomerular pathology or cellular infiltrates were found[29,30]. SARS-CoV was not detectable in kidneys from SARS-patients with AKI, suggesting that the AKI was related to multi-organ failure rather than SARS-CoV replication in the kidneys[31]. One possibility is that SARS-CoV-2 has a higher renal tropism than SARS-CoV, leading to increased affinity of SARS-CoV-2 for ACE2 and allowing for greater infection in kidneys. Studies have demonstrated that the SARS-CoV-2 virus can be transmitted by direct contact, respiratory droplets, fomites, and aerosols[32]. Moreover, viral RNA has been detected in the urine of some patients, and SARS-CoV-2 virus particles were also isolated from the urine of a patient with severe COVID-19[33]. Taken together, these results suggest that kidney may act as a viral reservoir, and kidney-originated viral particles may enter the urine through glomerular filtration.

The pathology of SARS-CoV-2 infection results from both direct and indirect injury, direct injury is caused by infection of the target cells by the SARS-CoV-2 virus, and indirect injury mainly results from immune responses, circulatory dysfunction, and hypoxia. In the kidney, ACE2 is highly expressed in the brush border of proximal tubular cells and, to a lesser extent, in podocytes, but not in glomerular endothelial and mesangial cells[16,34]. We here used immunohistochemistry and immunofluorescent double staining showed that both SARS-CoV-2 viral NP and S antigens accumulated in ACE2$^+$ kidney tubules (Fig. 3b). SARS-CoV-2 is a cytopathic virus that can also directly induce renal tubular injury during infection and replication.

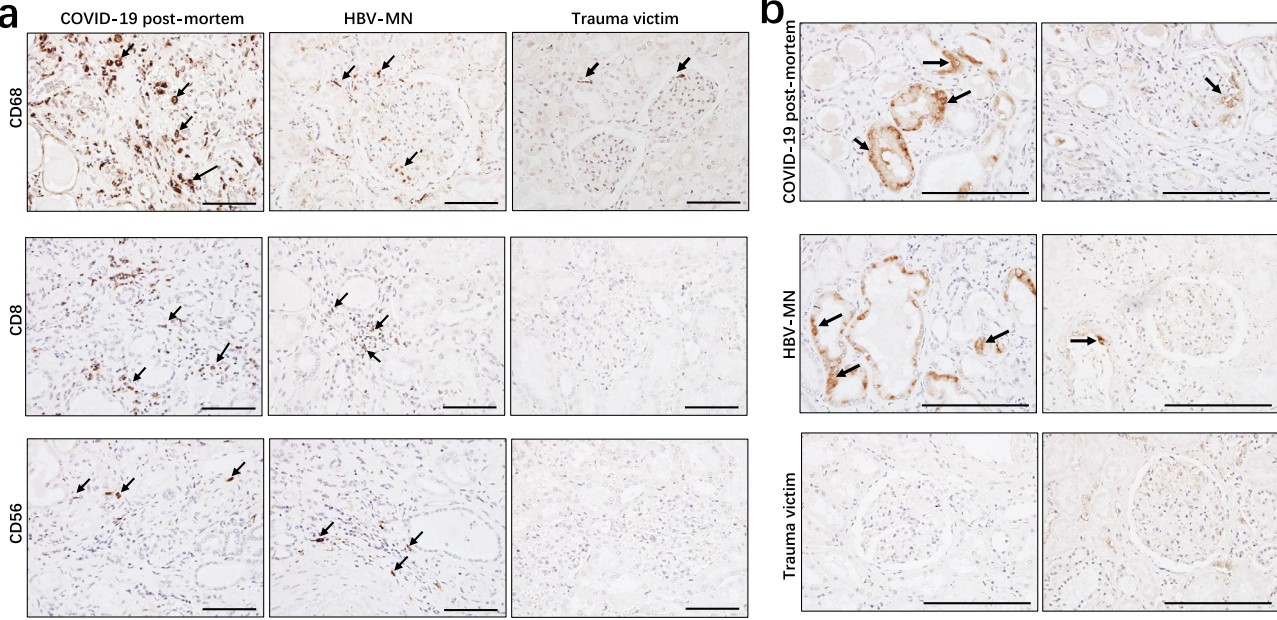

**Fig. 5 Immunohistochemistry analyzed leukocytes and the complement C5b-9 in kidney tissues.** Immunohistochemistry was used to detect the expression of **a** CD68, CD8, and CD56; **b** C5b-9 in kidney tissues from COVID-19 post-mortem (case #2), hepatitis B virus-associated membranous nephropathy (HBV-MN) and trauma victims. Arrow indicates positive cells. Scale bars = 200 μm. Data represent one of at least three technical replication each.

Interestingly, we found that SARS-CoV-2 NP and S antigens co-express with DP2 and PDS in kidney tubules, and the expression of DP2 and PDS was increased following SARS-CoV-2 infection (Fig. 4b), suggesting SARS-CoV-2 directly initiates hypoxic damage in infected kidney tubules. On the other hand, although host immune cells can infiltrate into infected tissue to counteract viral replication, hyperactivation of immune cells may instead promote fibrosis, induce epithelial cell apoptosis, and cause microvasculature changes[35–37]. We report here that the SARS-CoV-2 virus recruits high levels of CD68+ macrophages to infiltrate into the tubulointerstitium (Fig. 5a), suggesting that macrophages could induce tubular damage.

Complement is an integral component of the innate immune response and plays a key role in protective immunity against pathogens, but its excessive or deregulated activation may result in collateral tissue injury[38]. For example, the formation of C5b-9 can reduce the Zika virus load[39]. SARS-CoV can trigger the activation of complement component C3 and exacerbates the severity of SARS-CoV-mediated ARDS[40]. A recent study also reported that lung biopsy samples from patients with severe COVID-19 manifested C3a generation and C3-fragment deposition, suggesting SARS-CoV-2 activates the complementary system[35]. Moreover, both clinical and experimental models suggest that the abnormal presence of serum-derived complement components in the tubular lumen leads to the assembly of the complement C5b-9 on the apical brush border of tubular epithelial cells and that this is an important factor in the pathogenesis of tubulointerstitial damage[41–43]. We observed that SARS-CoV-2 initiates complement C5b-9 assembly and deposition on tubules (Fig. 5b). Similar to previous work[44], we also found sections from HBV-MN cases have C5b-9 deposition on tubules (Fig. 5b), suggesting that both SARS-CoV-2 and HBV can cause tubular damage through initiating C5b-9-mediated tubule pathogenesis. However, the mechanisms underlying how SARS-CoV-2 triggers C5b-9 formation are unclear. The complement can be activated through the classical, alternative and lectin pathways. The activation of the alternative pathway occurs by spontaneous hydrolysis of C3 in plasma. Critically ill COVID-9 patients also have very high serum concentrations of C3[35], suggesting that the alternative pathways can trigger C5b-9 formation.

We acknowledge that our study has several limitations. First, the number of patients in this study is relatively small, and the results should be validated by further studies. Second, there was no isolation of alive SARS-CoV-2 viruses from the kidneys due to the kidney samples being limited at that time. Third, electron microscopy (EM) and immuno-EM will be more definitive in validating the presence of SARS-CoV-2 in kidney tissues. In fact, we have performed these experiments; unfortunately, the results were not conclusive. Even though we detected some virus-like particles following EM, we could not exclude the possibility of sample contamination; due to the relatively low number of particles identified. Moreover, immuno-EM was performed in both lungs and kidneys, but the results were not satisfactory, and we were not able to detect viral particles in neither of the tissues. This was potentially due to the weakening of antigenicity after fixation by formalin, long-term preservation in paraffin wax at room temperature, and destruction of protein structure by ultrathin slicing.

In conclusion, we have demonstrated that the SARS-CoV-2 virus can directly infect human renal tubules and consequently lead to renal tubular injury and AKI, suggesting that the risk of AKI in COVID-19 patients should be kept in mind and alleviation of renal injury would benefit COVID-19 patients. Potential interventions such as blocking SARS-CoV-2/ACE2 binding, immune regulation and continuous renal replacement therapies (CRRT) for protecting kidney function in COVID-19 patients, particularly for AKI cases, are optional methods that may prevent fatalities.

## Methods

**Patients**. Medical records from 85 COVID-19 patients (aged from 21 years to 92 years) with dynamic observation of renal function in General Hospital of Central Theatre Command in Wuhan, China from January 17, 2020 to March 3, 2020 were collected and retrospectively analyzed. The kidney samples were obtained from

autopsies of six COVID-19 cases who were admitted in Jinyintan Hospital in Wuhan, China. Due to the special infection-control precaution of handling deceased subjects with COVID-19, post-mortem examination was performed in a designated pathology laboratory. The perished patients were taken post-mortem examination immediately (4 cases) or kept in −20 °C till postmortem within 24 h after death (2 cases). The kidneys from two trauma victims' autopsies (male, 65 years old; female, 62 years old) and two biopsies from HBV-related glomer-ulonephritis (HBV-MN, male, 59 years; female 60 years, both patients were pro-teinuric, and both were classified as stage I–II) were also involved. Diagnosis of COVID-19 was based on the New Coronavirus Pneumonia Prevention and Con-trol Program (5th edition) published by the National Health Commission of China. All the patients were laboratory-confirmed positives for SARS-CoV-2 by use of quantitative RT-PCR (qRT-PCR) on throat swab samples. This study was approved by the National Health Commission of China and Ethics Commission of General Hospital of Central Theatre Command ([2020]017-1) and Jinyintan Hospital (KY-2020-15.01). Due to the urgent need of treating this emerging epidemic, as well as the anonymization and retrospective analysis of clinical data, the written informed consent of the eighty-five patients with clinical data analyzed (shown in Table 1, Supplementary Table 1, Supplementary Fig. 1) were waived by the Ethical Com-mittee of General Hospital of Central Theatre Command under the terms of Ethical review of biomedical research involving human beings issued by National Health Commission of the People's Republic of China. The written informed consent of patients whose post-mortem tissues were collected was signed by their family before the autopsies.

**Data collection and definitions**. We reviewed clinical records and laboratory findings for all the patients. All information was obtained and curated with a customized data collection form. eGFR was calculated by the CKD-EPI equation based on serum creatinine level, sex, race, and age. AKI was defined as a decline of eGFR by at least 30% of the baseline value or below 90 ml/min. The classification of clinical types, which consist of mild/moderate/severe/critical, was based on the New Coronavirus Pneumonia Prevention and Control Program (5th edition) published by the National Health Commission of China. Three investigators (C. Wang, Z. Fen, and Y. Chen) directly collected the electronic medical records and independently reviewed the data collection forms to verify data accuracy. The other medical data collection of six patients whose post-mortem tissues were collected was executed by G. Wang and R. Wang.

**H&E staining**. Histopathological examination of kidneys from these autopsies or biopsy was performed by H&E staining. Briefly, paraffin-embedded tissue blocks were cut into 3 μm slices and mounted onto poly-lysine-coated glass slides. After dewaxing, tissues were incubated with hematoxylin for 5 min, after 1 min of dehydration by 100% alcohol, section were further treated with eosin for 30 s. Sections were mounted and the results were viewed using a light microscope (Zeiss Axioplan 2, Berlin, Germany).

**Immunohistochemistry**. The protocol for immunohistochemistry has been reported in our previous work[45]. Briefly, antigen retrieval was performed by microwaving these sections in citrate buffer (10 mM, pH 6.0). The sections were then incubated in 3% BSA plus 0.1% $H_2O_2$ for 1 h at RT to block nonspecific binding. The sections were then incubated overnight at 4 °C with primary anti-SARS-CoV-2 NP antibodies (clone ID: 019, 1:200, rabbit IgG; Sino Biological, Beijing), anti-SARS-CoV-2 NP antibodies (ab273434, 1:500, mouse monoclonal 6H3, Abcam), anti-SARS spike glycoprotein (S) antibodies (ab273433, 1:500, mouse monoclonal 1A9, Abcam), anti-ACE2 (clone ID: 10108-RP01, 1:100, rabbit IgG; Sino Biological), anti-CD8 (Clone ID:4B11, 1:100, mouse IgG2b; BIO-RAD), anti-CD68 (Clone ID:KP1, 1:100, mouse IgG1; BIO-RAD), anti-CD56 (Clone ID:123C3, 1:100, mouse IgG1; BIO-RAD), anti-C5b-9 (clone ID: aE11, 1:100, mouse IgG; Dako cytomation). Sections were further incubated with the Goat anti-Rabbit IgG (H + L) secondary antibody, HRP (#31460, Invitrogen) or Goat anti-Mouse secondary antibody, HRP (PA1-74421, ThermoFisher) for 1 h at RT, respectively. Peroxidase activity was visualized with the DAB Elite kit (K3465, DAKO), and the brown coloration of tissues represented positive staining as viewed by a light microscope (Zeiss Axioplan 2, Germany).

**Immunofluorescence double-staining**. For immunofluorescence double-staining, the sections were incubated with primary mouse originated antibodies including anti-SARS NP antibodies (ab273434, 1:500, mouse IgG), anti-SARS S antibodies (ab273433, 1:500, mouse IgG), or anti-DP2 (sc-271898, 1:200, mouse monoclonal C-5; Santa Cruz Biotechnology) and rabbit originated antibodies including anti-SARS NP antibodies (clone ID: 019, 1:200, rabbit IgG; Sino Biological, Beijing), anti-ACE2 (clone ID: 10108-RP01, 1:100, rabbit IgG; Sino Biological) or anti-PDS (ab182141, 1:100, rabbit IgG1; Abcam) at 4 °C overnight, respectively. After washing with PBS (3 washes, 5 min per wash), the sections were incubated with Alexa Fluor® 555-conjugated goat anti-rabbit IgG antibodies (Invitrogen, San Diego, CA, USA) and Alexa Fluor® 488-conjugated goat anti-rabbit IgG1 anti-bodies (Invitrogen) for an 1 h. Finally, the sections were incubated with 1 μg/ml DAPI (Sigma, St. Louis, MO, USA) for 10 min to stain the nuclei. Sections incu-bated with the appropriate isotype control primary antibodies and fluorescently labeled secondary antibodies were used as negative controls. The results were analyzed using fluorescence microscopy (Zeiss Axioplan 2).

**Detection of SARS-CoV-2 mRNA in kidneys by RNAscope**. RNA ISH was performed using RNAScope® Probe-V-nCoV2019-S (#848561, ACD, Newark, CA) directed against SARS-CoV-2 targeting 21631–23303 of NC_045512.2 and RNA-Scope® Probe-V-nCoV2019-S-sense (#845701, ACD). A negative control probe to the bacterial gene diaminopimelate B (DapB, #310043, ACD) was utilized to analyzed non-specific background, and a probe to the housekeeping gene pepti-dylprolyl isomerase B (PPIB) for RNA integrity (#313901, ACD) were used as positive control. The RNAscope® 2.5 HD Reagent Kit-RED (#322350, ACD) was utilized per the manufacturer's instructions as following. Briefly, the kidney tissue retrieval was performed at 95 °C for 20 min followed by incubation with RNAscope protease for 15 min at 40 °C. Probes were added and hybridized for 3 h at 42 °C, using the following protocol AMP1 3,3'-diaminobenzidine (DAB) (incubated for 30 min), AMP2 DAB (15 min), AMP3 DAB (30 min), AMP4 DAB (for 15 min), AMP5 DAB (30 min), and AMP6 DAB (15 min) were incubated followed by incubation with DAB for 20 min. Sections were counterstained with periodic acid-Schiff.

**Statistical analysis**. Statistical analyses were performed using GraphPad Prism version 8.0 (GraphPad Software, Inc., San Diego, CA, USA). Categorical variables were expressed as numbers (%). $p$ values are from $\chi^2$ (Table 1, Supplementary Table 1), two-side unpaired $t$ test (Table 1, Supplementary Table 1, Supplementary Fig. 1) and ordinary one-way ANOVA (Supplementary Fig. 1).

**Reporting summary**. Further information on research design is available in the Nature Research Reporting Summary linked to this article.

## Data availability
The authors declare that all data supporting the findings of this study are available within the article and its Supplementary Information Files or from the corresponding author upon request. Supplementary Fig. 1, Table 1, and Supplementary Table 1 have associated raw data. For the protection of patients' privacy, only anonymized data are available. The source files are provided within attached files. The other data associated with Figs. 1–5, Table 2, and Supplementary Figs. 2–4 are provided within the article and have not been included in the source data file. All the anonymized raw data could be provided from the corresponding authors upon request. Source data are provided with this paper.

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

## Acknowledgements

We are grateful for the support of patients and their family. This work was supported by The National Key Research and Development Program of China (2016YFA0502204), and the National Natural Science Foundation of China (NSFC; No. 81701551, 81971478, 81701562, and 81771691). The funding agencies did not participate in study design, data collection, data analysis, or manuscript writing. This manuscript has been released as a pre-print at MedRxiv, (Bo Diao et al. medRxiv 2020.03.04.20031120; 10.1101/2020.03.04.20031120).

## Author contributions

Yuzhang Wu and Yongwen Chen were involved in the final development of the project and manuscript preparation; Chenhui Wang, Zilin Yuan, Han Yang, Zeqing Feng, Ji Zhang, and Xiaotao Hou analyzed the data; Bo Diao, Chenhui Wang, and Huiming Wang performed most of experiments; Ying Liu, Gang Wang, Yinjun Tan, Han Yang, and Yueping Liu did H&E staining and immunohistochemistry; Changsong Wang evaluated H&E and immunohistochemistry results; Liang Liu, Rongshuai Wang, and Liang Ren provided COVID-19 autopsies and analyzed H&E staining.

## Competing interests

The authors declare no competing interests.
