## [Peer Review File · Nature Communications]

REVIEWER COMMENTS

Reviewer #1 (Remarks to the Author):

Diao and colleagues from Wuhan, China, describe the rates and risk factors associated with acute renal failure in 85 patients diagnosed with CoVID19 and wish to investigate whether there is direct contribution of SARS-CoV-2 (SCoV2) infection of renal tissues. Although the general observation linking severe cases of CoVID19 with multi-organ failure are like to be linked to critical illness and ARDS, especially in the elderly, the hypothesis is reasonable base on the reports of SCoV2 RNAemia and CoV-2ACE2 receptor expression in tubular epithelial cells. In this postmortem study (X=6, not mentioned in the abstract), severe acute tubular necrosis was seen in H&E stained tissues (Hypoxia? Postmortem?) and lymphocyte infiltration. IHC for the viral nucleocapsid protein/antigen using X was described as being positive, CD68+ macrophage infiltration, and C5b-9 complement deposition on tubules. The authors conclude that SCoV2 has direct cytotoxicity and activates inflammation and complement deposition, which induce acute renal failure.

Comments:

Major:

1. Data from 6 post-mortem were presented – reason for death? Time from death to post-mortem? What do control tissues of post-mortems deceased from other causes (septic shock, matched for time, age and co-morbidity look like? Is there hypoxic damage? Is the lymphocyte damage?

2. Table 1 provides a good overview, but has already been published, yet the number of deaths is missing as is the number of post-mortem performed, who had TEM studies done. Importantly, how many patients received medication for hypertension, and the type of anti-hypertensive medication is missing including the specific use of ACE blockers and -sartanes.

3. Table 2 is important, and should be expanded by adding duration in hospital, type and duration of respiration, time from death to postmortem, and the tissue viral load by PCR per reference gene.

4. P5: what was the clinical diagnosis for renal failure, and when did it occur in the course of the disease.

Please show representative large overview of cortex and medulla (incl 5- 10 glomeruli) for all cases, revealing whether or not, or to what extent nephrons were affected.

5. Please prepare a special comparison of case 6 versus 2, and case 4 versus 1, with appropriate comments about similarities and differences.

6. P6: the key message of this paper depends on Figure 2. However, the IHC shown is diffuse, and where the signals are strong, necrotic condensation seems to have taken place. It could represent unspecific protein casts. Immunofluorescence might provide more convincing data. The not infected controls appear very bleached. Again, juxtaposition of control postmortems passed away due to non SCoV2 causes are essential.

7. Quality control of anti-NP staining in infected respiratory cells need to be shown and preferably also from cell culture if the virus to compare patterns and intensities.

8. P6: Fig 3. Swollen mitochondria are expected due to hypoxemic and post-mortem damage, but not so nicely shown here (control postmortems? Only healthy probably biopsy tissue shown?). The wording virus-like particles is acceptable, but this does not look like CoV particles. What do the authors mean with "complete coating"?

9. Fig 4: (control postmortems? Only healthy probably biopsy tissue shown?).

10. The discussion is very general and superficial, focusing mainly on the assumed demonstration of their interpretation of the data show.

The expression "cytokine storm" is very popular and widely used, as it appears to aim at generating general consent to "we don't know what is going on". It would be important to leave this suspicious term and come forward with more fact-based information.

Here, extension to other coronaviruses would be in order.

Specifically, why is a primary infection with this virus not causing viremia and acute kidney failure in many of the younger adults?

11. What is the difference to viruses that are known to directly infect the kidney during primary viremia?

12. What is activating the complement MAC and is that not also seen in another pathologies (see

above)?

13. What do these authors believe, are important parameters to be taken into account for performing an informative study on this topic, such that other groups can give reference to this study.

Minor:

1. Intro: Duplicate abbreviation for SARS-CoV-2 twice
2. Intro: proteinuria is unspecific – glomerular or tubular proteinuria profiles?
3. Intro: Alternative cause or causes in the patients not presented, please add.
4. Results: Proteinuria profiles - consistent with tubular necrosis? Many clinical tests only use albumin, please comment.
5. Statistics: please be more generous in explaining the statistics, one-or two-sides P-values etc...

Reviewer #2 (Remarks to the Author):

To Authors

This postmortem histopathology study has provided data on the potential impact of SARS-CoV-2 (COVID-19) on human kidney. The authors present data and sample collection analyses of renal data including eGFR calculations along with other clinical parameters from 85 patients with laboratory-confirmed COVID-19 admitted to a hospital in Wuhan in early 2020, in that 27% (23/85) patients developed AKI (acute kidney injury). The authors report that COVID-19 infection can be associated with AKI as a result of direct renal tubular cytotoxicity. If these data are correct, this can be considered an important discovery.

1. The claim that AKI is the result of virus directly attacking and injuring kidney tubules leading to ATN and not secondarily due to other concurrent infections is a major discovery that needs to be better substantiated. Were the viral particles in kidneys consistent across most sections?
2. One limitation is that kidney tissues were from six patients with postmortem examinations, but if data are exceptionally consistent and reproducible, then there is an acceptable proof of concept.
3. Did H&E staining show coronavirus specific patterns of injury, and if so in which unique ways? H&E staining showed severe acute tubular necrosis (ATN) and lymphocyte infiltration, which is very interesting but non-specific.
4. THR authors report that they examined in situ expression of viral nucleocapsid protein antigen, immune cell markers including CD8, CD68 and CD56, and complement C5b-9, which were all detected by immune-histochemistry. How specific are these markers and what are the chance of having these from other concurrent events?
5. A supportive finding is that the virus initiates CD68+ macrophage together with complement C5b-9 deposition that is a biologically plausible mechanism to mediate tubular pathogenesis, but this would benefit from an animal model or some in vitro experiment.
6. Can the authors measure Prostaglandin D2 and the complex PGD2/DP2 in the tissues as a potential pathophysiologic pathway?
7. The authors need to think of similar nephrotoxicity models from other familiar viruses such as influenza virus to cause direct renal tubules
8. The statement that the virus initiates CD68+ macrophage together with complement C5b-9 deposition to mediate tubular pathogenesis is a strong statement and needs more supportive data.

Minor comments:

9. The authors state that the elderly patients and those with comorbidities especially hypertension and heart failure were more likely to develop AKI and then they say 65% vs 24%, 70% vs 11%, respectively. What groups do these numbers refer to? This is an example that the text needs improvement
10. The conclusion statement need to be more accurate e.g. "COVID-19 infection can be associated with AKI and that in some (or most?) of these cases the AKI event from direct renal tubular cytotoxicity as the result of virus directly attacking an injuring kidney tubules leading to ATN and not secondarily due to other concurrent infections.

11. Use no decimal for percentages > 10%, e.g. 27.06% should be 27%.
12. Replace acute renal failure (ARF) with acute kidney injury (AKI)
13. The English language is poor and needs major improvement.

Reviewer #3 (Remarks to the Author):

This manuscript comprises a retrospective analysis of renal failure potentially associated with kidney infection in 85 hospitalized COVID19 patients. Acute renal failure assessed by eGFR was observed in 27% of patients, predominantly in elderly individuals and those with comorbidities. The involvement of kidney disease has been noted in other COVID19 studies. What makes this report novel is that it provides direct evidence for direct SARS-CoV-2 infection of kidneys using histological analysis of autopsy material from 6 post mortem cases. Evidence for viral infection is provided by both immunohistochemistry for viral antigen (NP) in all 6 samples as well as by EM in two samples. Results using H&E staining as well as staining for CD68, CD8, CD56 and C5b-9 deposition further revealed benign glomerulosclerosis, but severe acute tubular necrosis associated with strong NP expression specifically in tubules in all 6 cases. However, viral antigen did not appear to correlate with interstitial leukocyte infiltration, as inflammation was only severe in two cases, modest in 3 and not detected in one case. Similarly, although infiltrates appeared to be dominated by CD68+ macrophages over CD8 and NK cells, relative abundance varied; a correlation of virus load with complement deposition was also not evident in all cases. Conclusion drawn in the abstract, lines 141, 169/170 and 176/177 suggesting tubular necrosis is attributed to macrophages and/or complement deposition thus appear tentative and should be tempered.

Several additional experiments and clarifications would significantly strengthen the manuscript:

1. ACE2 expression in kidneys should be shown by staining, as reference is made to two pre-reviewed manuscripts (ref 11,12) suggesting ACE2 is expressed in kidneys. However, actual demonstration of ACE2 expression in healthy kidneys as well as ACE2 expression in relationship to viral NP in injured tissues would be informative to reveal a correlation.

2. It is indicated that normal kidney tissues and renal sections from 'unrelated' autopsies were used for secondary control Ab staining. Staining of the COVID19 patient autopsy material itself needs to be tested for reactivity of rabbit Ab, as the inflammation may enhance unspecific reactivity. (It is also not clear what unrelated autopsies refers to).

3. Fig. 2. Interpretation of the viral antigen stain should be explained. What is the determination of a viral inclusion body based on, as NP+ cells in the lung are much smaller than areas stained in Kidney? The resolution also appears insufficient to substantiate the interpretation that no nuclear NP staining was observed (L147).

4. Table 2: It would be helpful if grading of 1+ to 4+ can be quantified by positive staining area or cell numbers, as the distinction as provided is somewhat arbitrary. Was more than one section stained ?

Minor issues:

1. It should be made clear if the case numbers in Fig. 1 correlate with the Table. Figures 2 and 3 should also indicate the case numbers they represent.

2. Several English grammar/vocabulary concerns should be corrected:

Line 75- 'biopsies' should be changed to 'autopsies'

Table 2: Interstitial rather than 'Intestinal' ?

Point to point response to reviewers' comment

Reviewer #1 (Remarks to the Author):

Diao and colleagues from Wuhan, China, describe the rates and risk factors associated with acute renal failure in 85 patients diagnosed with CoVID19 and wish to investigate whether there is direct contribution of SARS-CoV-2 (SCoV2) infection of renal tissues. Although the general observation linking severe cases of CoVID19 with multi-organ failure are like to be linked to critical illness and ARDS, especially in the elderly, the hypothesis is reasonable base on the reports of SCoV2 RNAemia and CoV-2 ACE2 receptor expression in tubular epithelial cells. In this postmortem study (X=6, not mentioned in the abstract), severe acute tubular necrosis was seen in H&E stained tissues (Hypoxia? Postmortem?) and lymphocyte infiltration. IHC for the viral nucleocapsid protein/antigen using X was described as being positive, CD68+ macrophage infiltration, and C5b-9 complement deposition on tubules. The authors conclude that SCoV2 has direct cytotoxicity and activates inflammation and complement deposition, which induce acute renal failure.

A: We appreciate the reviewer's criticism. The abstract has been revised. The co-expression of SARS-CoV-2 NP antigen and ACE2 on tubular epithelial cells were also detected by immunofluorescence staining (Figure 4). Hopefully these modifications are adequate.

Comments:

Major:

Q1.Data from 6 post-mortem were presented – reason for death? Time from death to post-mortem? What do control tissues of post-mortems deceased from other causes (septic shock, matched for time, age and co-morbidity look like? Is there hypoxic damage? Is the lymphocyte damage?

A1: We appreciate the reviewer's comments. The reasons of deaths of these 6 post-mortem were acute respiratory distress syndrome (ARDS) caused by SARS-CoV-2 infection. Two bodies were kept in -20°C for one day till post-mortem, whereas, other four cases were done post-mortem immediately. All of these 6 cases have lymphocytopenia, suggesting that they have lymphocyte damage in circulation. Here we also collected kidney tissues from two cases of trauma victims (male, 65 years old; female, 62 years old) and two biopsies from HBV- related glomerulonephritis (HBV-MN, male, 59 years; female 60 years, both patients were proteinuric, and both were classified as stage I-II). These samples were used as controls. No obvious lymphocyte infiltration, tubular or glomerular injury was observed in sections from trauma victims, as detected by H&E staining (Figure 1). The histopathology of kidney from hepatitis B virus-associated membranous nephropathy (HBV-MN) patients, on the other hand, the glomerulus was segmental necrosis or segmental sclerosis, whereas, the renal tubules manifested quite normal and no obvious atrophy or necrosis were observed (Figure 1). Collectively, it seems that SARS-CoV-2 mediated kidney histopathology is different to nephritis and nephrotic syndrome, in which the

pathological signs are mainly shown in glomerulus, rarely with tubular injury. These signs are also different from toxic tubular injury, which manifest exfoliation and necrosis of tubular cells, interstitial edema but not lymphocytic infiltration. What's more, ischemic ATN have damage in all renal tubules , but basement membrane damage is common, which is rarely found in these six autopsies.

Immunofluorescent staining showed that the expression of hypoxic damage molecules, including DP2 and PDS, in COVID-19 kidney tubules was induced, and both members were co-expressed with SARS-CoV-2 NP protein, suggesting SARS-CoV-2 virus can directly initiate hypoxic damage (Figure 5). Hopefully these additional experimental data are acceptable.

Q2: Table 1 provides a good overview, but has already been published, yet the number of deaths is missing as is the number of post-mortem performed, who had TEM studies done. Importantly, how many patients received medication for hypertension, and the type of anti-hypertensive medication is missing including the specific use of ACE blockers and -sartanes.

A2: The novel overview data, the death rate, the medications used, clinical manifestations and classification of clinical types were also added in Table 1. Moreover, cases that were done with TEM observation were indicated in Table 2 of revised manuscript. In these study patients, only a part of them used ACEI and ARB, making it hard to evaluate the side function of these anti-

hypertensive medications. It seems that ACEI/ARB treatment does not lead to AKI (Table 1). Recently, a study has reported that the anti-hypertensive medications including Angiotensin Converting Enzyme Inhibitors (ACEI) and Angiotensin II Receptor Blockers (ARB) did not affect the mortality of patients with hypertension hospitalized with COVID-19 (Zhang P, et al., PMID: 32302265), suggesting that use of ACEI/ARB was not associated with an increased mortality risk among hospitalized COVID-19 patients with hypertension.

Q3: Table 2 is important, and should be expanded by adding duration in hospital, type and duration of respiration, time from death to postmortem, and the tissue viral load by PCR per reference gene.

A3: Two bodies were kept in -20°C for one day till post-mortem, whereas, other four cases were done post-mortem immediately. The duration in hospital, type and duration of respiration, time from death to postmortem of each case were added in the revised Table 2. However, we have not the capacity to detect SARS-CoV-2 viral gene by PCR because we did not have P3 level lab to operate this test.

Q4: P5: what was the clinical diagnosis for renal failure, and when did it occur in the course of the disease. Please show representative large overview of cortex and medulla (incl 5- 10 glomeruli) for all cases, revealing whether or not, or to what extent nephrons were affected.

A4: The clinical diagnosis for renal failure has been mentioned in METHOD. AKI was defined as a decline of eGFR by at least 30% of the baseline value or below 90 ml/min. The renal failure from disease onset is added in Table 1. It is shown that the AKI occurred in the early stage of COVID-19. The representative large overview of cortex and medulla for all cases were also added (Figure 1). Hopefully they are adequate.

Q5: Please prepare a special comparison of case 6 versus 2, and case 4 versus 1, with appropriate comments about similarities and differences.

A5: In the revised manuscript, we compared case 6 versus 2, and case 4 versus 1. Please check it in the main text, for your convenience, I put it here as well. To further understand the influence of age and coexisting disorder on AKI, we compared two pairs of autopsies (#2 vs. #6 and #1 vs. #4). The kidney sections from case #2 , a 86 years old female patient with hypertension and coronary disease for more than 10 years, manifested obvious glomerulosclerosis, protein accumulation within the tubules, severe interstitial fibrosis, along with acute pathological signs such as tubular epithelial cells exfoliation, necrosis, vacuolation, strong lymphocytes interstitial infiltration, whereas sections from case #6 , a 67 years old female patient without coexisting disorders, showed scarcely glomerular changes and milder luminal brush border sloughing and tubular necrosis. Correspondingly, case #1, a 70 years old male patient with hypertension for more than 10 years, whereas #4, a 51 years old male patient

with hypertension for over 10 years, both have glomerulosclerosis, luminal brush border sloughing, epithelial cells exfoliation, necrosis, lymphocytes interstitial infiltration, although the signs case #4 presented are slightly milder than those in case #1 (Fig. 1, Table 2). All these results suggest that age and coexisting disorders both could exacerbate the renal injury. It seems that coexisting disorders are the more serious risk factor. All the glomerular changes are more likely attributed to underlying diseases.

Q6: P6: the key message of this paper depends on Figure 2. However, the IHC shown is diffuse, and where the signals are strong, necrotic condensation seems to have taken place. It could represent unspecific protein casts. Immunofluorescence might provide more convincing data. The not infected controls appear very bleached. Again, juxtaposition of control postmortems passed away due to non SCoV2 causes are essential.

A6: We appreciate this valuable comments. The expression of SARS-CoV-2 specific NP protein and ACE2 in kidney tissues from autopsies and controls was detected by immunofluorescence staining, and results were added in Figure 4b. The SARS-CoV-2 viruses- like particles were also seen in kidney samples by TEM. Moreover, the kidneys from two trauma victims' autopsies (male, 65 years old; female, 62 years old) and two biopsies from HBV- related glomerulonephritis (HBV-MN, male, 59 years; female 60 years, both patients were proteinuric, and both were classified as stage I-II) were also involved. All of

these tissues and isotype antibodies were set up as negative controls which could exclude unspecific reactivity. Collectively, these combined data confirmed that SARS-CoV-2 exist in kidneys from COVID-19 autopsies.

Q7: Quality control of anti-NP staining in infected respiratory cells need to be shown and preferably also from cell culture if the virus to compare patterns and intensities.

A7: We have not the capacity to culture SARS-CoV-2 virus in vitro due to the P3 level lab is not available in here. Here the expression of SARS-CoV-2 NP antigen in lung tissues from COVID-19 autopsy was used as positive control (Supplementary Figure 2). Quality control has been performed by Antibody provider as well.

Q8: P6: Fig 3. Swollen mitochondria are expected due to hypoxemic and post-mortem damage, but not so nicely shown here (control postmortems? Only healthy probably biopsy tissue shown?). The wording virus-like particles is acceptable, but this does not look like CoV particles. What do the authors mean with “complete coating”?

A8: Figure 3 was revised, a clearer figure with higher resolution was added, and swollen mitochondria were obvious. We did not have control postmortems to do TEM observation. The “complete coating” was deleted in the revised manuscript. We agree with the reviewer that the wording virus-like particles may be acceptable due to the diameter of this particle. However, the crown on the

surface of this virus was not clear enough but still can be seen if this figure was enlarged, which was why we chose to use the wording ‘coronavirus- like particles’.

Q9: Fig 4: (control postmortems? Only healthy probably biopsy tissue shown?).

A9: Figure 4 was revised, the control sections from trauma victims and HBV-MN were added. Thank you very much!

Q10: The discussion is very general and superficial, focusing mainly on the assumed demonstration of their interpretation of the data show. The expression “cytokine storm” is very popular and widely used, as it appears to aim at generating general consent to “we don’t know what is going on”. It would be important to leave this suspicious term and come forward with more fact-based information. Here, extension to other coronaviruses would be in order. Specifically, why is a primary infection with this virus not causing viremia and acute kidney failure in many pf the younger adults?

A10: We appreciate the reviewer's comments. The discussion part was re-written and more references were added in the revised manuscript, such as data from other coronaviruses including SARS-CoV and MERS-CoV were also recruited in the revised manuscript.

Lymphocytes play an essential role in fighting against viral infections, therefore, boosting the number and enhancing the antiviral function of T cells in

COVID-19 patients is of paramount value for successful recovery. However, results of other researches and our work have showed that most COVID-19 cases displayed severe lymphocytopenia, especially in aged cases. However, the younger adults have very high numbers of lymphocytes in circulation and these cells can successfully and timely clear viruses. Further, lymphocytopenia is believed to be the result of cytokine storm, and the continuous low-level inflammation which usually exists in the elder people may be the major reason to induce cytokine storm. So, we think a primary infection with this virus does not cause viremia and acute kidney injury in many of the younger adults.

Q11: What is the difference to viruses that are known to directly infect the kidney during primary viremia?

A11 : In the new version, we compared SARS-CoV-2 with Hantaan virus (Hanta virus, HTNV). Actually, it seems that SARS-CoV-2 mediated kidney histopathology is similar to HTNV infection, which induces acute tubulointerstitial nephritis following the infiltration of inflammatory cells. However, the vascular alterations and leakage are the major cause of HTNV-associated AKI, and this is less founded in the COVID-19 kidneys. By the way, some other kidney injuries are also involved. Hopefully they are adequate.

Q12: What is activating the complement MAC and is that not also seen in another pathologies (see above)?

A12: Although we here did not know how SARS-CoV-2 activated MAC in

kidney tubules, several studies have reported that some viruses can activate the complement alternative pathway to trigger C3 release, thus lead to MAC (C5b-9) deposition in tissues. For example, MAC plays a role in fulminant hepatitis. We therefore conjugate that SRAR-CoV-2 might also use the similar pathway to activate MAC. We found that HBV-MN can cause MAC deposition in renal glomerulus as well. We also discussed this point in the discussion part. Hopefully these modifications are adequate.

Q13: What do these authors believe, are important parameters to be taken into account for performing an informative study on this topic, such that other groups can give reference to this study.

A13: The most important parameter is that we confirmed that SARS-CoV-2 virus can directly infect human kidneys and the renal tubular epithelial cells are the targets of SARS-CoV-2 virus. The most important parameters that should be taken are serum creatinine and urea levels, urinary protein (including microglobulin) and urine glucose levels. These would help to monitor renal function, especially renal tubular function.

Minor:

Q1: Intro: Duplicate abbreviation for SARS-CoV-2 twice.

A1: We have revised it. Thank you very much!

Q2: Intro: proteinuria is unspecific – glomerular or tubular proteinuria profiles?

A2: The glomerular proteinuria may have a large protein leak, whereas, the renal tubular is reabsorption disorder, and may have microalbuminuria. However, we do not have the capacity to differentiate glomerular or tubular proteinuria profiles based on the available data. We discussed this issue in the A4 as well.

Q3: Intro: Alternative cause or causes in the patients not presented, please add.

A3: We collected all the patients with dynamic observation of renal function in General Hospital of the Central Theatre Command in Wuhan from January 17, 2020 to February 13, 2020. Other patients in this hospital did not have dynamic observation, so that we could not confirm they had AKI or not, which makes them excluded from this study.

Q4: Results: Proteinuria profiles - consistent with tubular necrosis? Many clinical tests only use albumin, please comment.

A4: In the case of mild renal tubular injury, it is manifested as reabsorption disorder, with the increase of urinary α or β microglobulin. However, the patients in this study have no relevant data. On the one hand, doctors did not realize the risk of renal tubular injury in the early stage of the epidemic. On the other hand, these tests could not be performed through automatic instrument, and it increased the risk of infection for doctors. Comparing the data of AKI and

non AKI, we found that the differences of urine glucose or electrolyte disorder were consistent with tubular necrosis. It should be noted that glomerular injury is often associated with severe tubular injury, and the performance of renal failure is more serious at this stage, glomerular function tests (such as urinary albumin) are still quite necessary.

Q5: Statistics: please be more generous in explaining the statistics, one-or two-sides P-values et.

A5: Thank you for this suggestion. Statistical analysis has been revised as suggested. Because of it's a newly emerged disease that we know scarcely before, p-values are from two-sides comparisons to avoid the interference of subjective factors.

Point to point response to reviewers' comment

Reviewer #2 (Remarks to the Author):

To Authors:

This postmortem histopathology study has provided data on the potential impact of SARS-CoV-2 (COVID-19) on human kidney. The authors present data and sample collection analyses of renal data including eGFR calculations along with other clinical parameters from 85 patients with laboratory-confirmed COVID-19 admitted to a hospital in Wuhan in early 2020, in that 27% (23/85) patients developed AKI (acute kidney injury). The authors report that COVID-19 infection can be associated with AKI as a result of direct renal tubular cytotoxicity. If these data are correct, this can be considered an important discovery.

Q1. The claim that AKI is the result of virus directly attacking and injuring kidney tubules leading to ATN and not secondarily due to other concurrent infections is a major discovery that needs to be better substantiated. Were the viral particles in kidneys consistent across most sections?

A1: First of all, thank you so much for the positive remarks of our study. In Figure 3, most tubules in sections from case #2 and Case #6 have SARS-CoV-2 NP antigen. Two kidney tissues were also used to detect viruses by TEM, and results showed both cases have virus- like particles. Collectively, these results SARS-CoV-2 directly infect human kidney tubules.

Q2. One limitation is that kidney tissues were from six patients with postmortem examinations, but if data are exceptionally consistent and reproducible, then there is an acceptable proof of concept.

A2: We totally agree with the limitation pointed out by this reviewer. We have put this limitation in our Discussion section. Generally, the more patients involved in postmortem examinations, the more convincing conclusions we can draw. However, only six patients with postmortem examinations were available at early February. Additionally, the Sections from all of these six COVID-19 autopsies manifested acute kidney injury. Thirdly, the SARS-CoV-2 specific NP antigen was seen in tubules of all these 6 cases. Collectively, this was why we drew this conclusion in our manuscript.

Q3. Did H&E staining show coronavirus specific patterns of injury, and if so in which unique ways? H&E staining showed severe acute tubular necrosis (ATN) and lymphocyte infiltration, which is very interesting but non-specific.

A3: Results showed that varying degrees of tubular necrosis, luminal brush border sloughing, vacuole degeneration and lymphocyte infiltration were found in these six renal specimens. Severe glomerular injury was absent in all cases, although benign hypertensive glomerulosclerosis was noted in four hypertensive patients (case #1, #2, #4 and #5). No obvious lymphocyte infiltration in the tubulointerstitium, tubular damage or glomerular injury was observed in kidney sections from trauma victims (Fig. 1). The histopathology of kidney from

hepatitis B virus-associated membranous nephropathy (HBV-MN) patients. On the other hand, the glomerulus was segmental necrosis or segmental sclerosis, whereas, the renal tubules manifested quite normal and no obvious atrophy or necrosis were observed (Fig. 1). Actually, it seems that SARS-CoV-2 mediated kidney histopathology is different to nephritis and nephrotic syndrome, in which the pathological signs are mainly shown in glomerulus, rarely with tubular injury. This signs are also different from toxic tubular injury, which manifest exfoliation and necrosis of tubular cells, interstitial edema but not lymphocytic infiltration. Moreover, ischemic acute tubular necrosis (ATN) has damage in all renal tubules, but basement membrane damage is common, which is rarely found in these six autopsies.

Q4. THR authors report that they examined in situ expression of viral nucleocapsid protein antigen, immune cell markers including CD8, CD68 and CD56, and complement C5b-9, which were all detected by immune-histochemistry. How specific are these markers and what are the chance of having these from other concurrent events?

A4: We did the IHC as well as IF to confirm the expression of this markers. These markers are so specific that are widely used to validate the immune cell infiltration or immune injury. It could represent unspecified protein casts in some cases. In order to exclude the chances of having these from other concurrent events, the control tissues and the isotype control were enrolled in

our study, whose results were completely negative.

Q5. A supportive finding is that the virus initiates CD68⁺ macrophage together with complement C5b-9 deposition that is a biologically plausible mechanism to mediate tubular pathogenesis, but this would benefit from an animal model or some in vitro experiment.

A5: Due to the deficiency of great animal model and P3 condition, we are sorry that we cannot do such experiments to prove this mechanism at present. Profound thanks for this remarkable suggestion, we may continue this work in the future if possible.

Q6. Can the authors measure Prostaglandin D2 and the complex PGD2/DP2 in the tissues as a potential pathophysiologic pathway?

A6: Thanks for this splendid advice, which we think will make the conclusions of the revised manuscript more robust. We measured the expression of PDS and DP2 by the immunofluorescent staining (the PGD2 antibodies are not available till now). Please kindly check the data in the attached files (Figure 5). Results showed that both of them represented a high expression in kidney tubules from COVID-19 autopsies, and they were co-expressed with SARS-CoV-2 specific NP protein (Fig. 5a, b). Conversely, sections from trauma victims have not any DP2 and only manifested low level of PDS (Fig. 5c), suggesting SARS-CoV-2 virus can directly initiate hypoxic damage in infected kidney tubules.

Q7. The authors need to think of similar nephrotoxicity models from other familiar viruses such as influenza virus to cause direct renal tubules.

A7: In the new version, we compared SARS-CoV-2 with Hantaan virus (Hanta virus, HTNV). Actually, it seems that SARS-CoV-2 mediated kidney histopathology is similar to HTNV infection, which induces acute tubulointerstitial nephritis following the infiltration of inflammatory cells. However, the vascular alterations and leakage are the major cause of HTNV-associated AKI, and this is less founded in the COVID-19 kidneys. By the way, some other kidney injuries are also involved. Hopefully they are adequate.

Q8. The statement that the virus initiates CD68+ macrophage together with complement C5b-9 deposition to mediate tubular pathogenesis is a strong statement and needs more supportive data.

A8: We totally agree with this comment. Macrophages are the most common infiltrating cells in viral infection, which is consistent with our observation. In the revised manuscript, strong C5b-9 depositions on tubules were observed in all of these six cases, and two out of the six cases manifested with low levels of C5b-9 deposition on glomeruli and capillaries (Fig. 6b, Table 2). However, C5b-9 deposition is absent in sections from trauma victims, whereas, sections from HBV-MN also manifested high levels of C5b-9 deposition in both tubules (Fig. 6b), suggesting both SARS-CoV-2 and HBV can induce kidney tubular lesion through triggering C5b-9 deposition. This is consistent with other infection

models in which MAC can cause disease damage. However, we do not have P3 laboratory conditions for now, so the validation using animal models and intervention experiments cannot be performed .

Minor comments:

Q9. The authors state that the elderly patients and those with comorbidities especially hypertension and heart failure were more likely to develop AKI and then they say 65% vs 24%, 70% vs 11%, respectively. What groups do these numbers refer to? This is an example that the text needs improvement

A9: We revised THE WHOLE TEXT to improve the accuracy.

Q10. The conclusion statement need to be more accurate e.g. “COVID-19 infection can be associated with AKI and that in some (or most?) of these cases the AKI event from direct renal tubular cytotoxicity as the result of virus directly attacking an injuring kidney tubules leading to ATN and not secondarily due to other concurrent infections.

A10: We revised conclusion statement , please check it in the main text.

Q11. Use no decimal for percentages > 10%, e.g. 27.06% should be 27%.

A11: Thank you for the advice. We have revised them.

Q12. Replace acute renal failure (ARF) with acute kidney injury (AKI)

A12: Thank you for the advice. We have revised them.

Q13. The English language is poor and needs major improvement.

A13: We appreciate your frank comments and have invited a native English speaker to revise our manuscript.

Point to point response to reviewers' comment

Reviewer #3 (Remarks to the Author):

This manuscript comprises a retrospective analysis of renal failure potentially associated with kidney infection in 85 hospitalized COVID19 patients. Acute renal failure assessed by eGFR was observed in 27% of patients, predominantly in elderly individuals and those with comorbidities. The involvement of kidney disease has been noted in other COVID19 studies. What makes this report novel is that it provides direct evidence for direct SARS-CoV-2 infection of kidneys using histological analysis of autopsy material from 6 post mortem cases. Evidence for viral infection is provided by both immunohistochemistry for viral antigen (NP) in all 6 samples as well as by EM in two samples. Results using H&E staining as well as staining for CD68, CD8, CD56 and C5b-9 deposition further revealed benign glomerulosclerosis, but severe acute tubular necrosis associated with strong NP expression specifically in tubules in all 6 cases. However, viral antigen did not appear to correlate with interstitial leukocyte infiltration, as inflammation was only severe in two cases, modest in 3 and not detected in one case. Similarly, although infiltrates appeared to be dominated by CD68+ macrophages over CD8 and NK cells, relative abundance varied; a correlation of virus load with complement deposition was also not evident in all cases. Conclusion drawn in the abstract, lines 141, 169/170

and 176/177 suggesting tubular necrosis is attributed to macrophages and/or complement deposition thus appear tentative and should be tempered.

Several additional experiments and clarifications would significantly strengthen the manuscript:

Q1. ACE2 expression in kidneys should be shown by staining, as reference is made to two pre-reviewed manuscripts (ref 11,12) suggesting ACE2 is expressed in kidneys. However, actual demonstration of ACE2 expression in healthy kidneys as well as ACE2 expression in relationship to viral NP in injured tissues would be informative to reveal a correlation.

A1: We totally agree with this comment. In the revised manuscript, immunohistochemistry showed that kidney tissues from COVID-19 post-mortem have ACE2 expression (Fig. 4a). Immunofluorescent double staining illustrated that SARS-CoV-2 NP antigen are colocalized to ACE2 protein, and the expression is restricted in kidney tubules (Fig. 4b). The sections from HBV-MN and trauma victims' biopsies are also positive for ACE2, however, these sections are absent in these sections. Collectively, these data suggest that SARS-CoV-2 might directly infect human kidney tubules through ACE2. Hopefully these additional data are adequate.

Q2. It is indicated that normal kidney tissues and renal sections from 'unrelated' autopsies were used for secondary control Ab staining. Staining of the COVID19

patient autopsy material itself needs to be tested for reactivity of rabbit Ab, as the inflammation may enhance unspecific reactivity. (It is also not clear what unrelated autopsies refers to).

A2: In the revised manuscript, the kidneys from two trauma victims' autopsies (male, 65 years old; female, 62 years old) and two biopsies from HBV- related glomerulonephritis (HBV-MN, male, 59 years; female 60 years, both patients were proteinuric, and both were classified as stage I-II) were also involved. These sections were used as control. Moreover, sections incubated with isotype antibodies as the negative control which could exclude unspecific reactivity. Thank you very much!

Q3. Fig. 2. Interpretation of the viral antigen stain should be explained. What is the determination of a viral inclusion body based on, as NP+ cells in the lung are much smaller than areas stained in Kidney? The resolution also appears insufficient to substantiate the interpretation that no nuclear NP staining was observed (L147).

A3: In the revised manuscript, the expression of SARS-CoV-2 NP antigen in lung tissues was moved to supplementary Figure 2. In the lung tissue, SARS-CoV-2 infected type □ alveolar epithelial cells, which are smaller than kidney tubular epithelial cells. Viral inclusion body is dense, bulky virus crystal , which is shown in the Fig3(case#2, triangle arrow). It is aggregation of insoluble proteins and RNA , so that can be stained by NP-protein antibody. In figure 3,

higher resolution of figures was showed and results demonstrated that NP antigen is absence in tubular nuclear.

Q4. Table 2: It would be helpful if grading of 1+ to 4+ can be quantified by positive staining area or cell numbers, as the distinction as provided is somewhat arbitrary. Was more than one section stained?

A4: Definition of grading of 1+ to 4+ is updated in the revised manuscript. For example, -: negative, +<10% tubules are positive; ++ 10%~ 40% tubules are positive; +++ 40%~ 60% tubules are positive; ++++ >60% tubules are positive. We stained a couple of sections each tissue specimen and the majority of them are identical. Thank you very much!

Minor issues:

Q1. It should be made clear if the case numbers in Fig. 1 correlate with the Table Figures 2 and 3 should also indicate the case numbers they represent.

A1: Thank you for this kind advice, we indicated them in the method, result section and figure/table legends.

Q2. Several English grammer/vocabulary concerns should be corrected:

Line 75- 'biopsies' should be changed to 'autopsies'

Table 2: Interstitial rather than 'Intestinal' ?

A2: Thanks for your careful collating, we have rectified them.

REVIEWER COMMENTS

Reviewer #2 (Remarks to the Author):

Well done

Reviewer #3 (Remarks to the Author):

This revised manuscript addressed some concerns raised by the previous review, including more extensive data on the patient cohort, preexisting conditions, treatment, etc. The main novelty continues to reside in evaluation of kidney autopsy material from 6 post mortem cases linking viral antigen expression with tubule damage. New data is provided with respect to COVID19 and control tissues, as well as expression of viral antigen in relation to ACE2 and DP2/PDS to provide a correlation between location of viral antigen and tissue damage. However, there are ongoing and new concerns with respect to nucleocapsid antigen staining, costaining, and interpretation with respect to leukocyte infiltrates and complement deposition, leaving conclusions tentative.

Major ongoing concerns:

1. Time between autopsy and death should be included in Methods as long interval may affect results.
2. Fig. 2 only shows EM for the same section of one patient sample (not 2 patient samples as indicated) and the 500 nm Figures appear the same. It is still unclear what structural parameters were chosen for the arrow highlighted densities termed as virus like.
3. The nucleocapsid stain in Fig. 3 remains fuzzy and not very specific despite the negative controls. The interpretation of a viral inclusion body also remains tentative as it appears too large. Have the authors evaluated other anti N or S antibodies to confirm staining patterns?
4. Figures 4 and 5 use primary anti N, ACE2, DP2 and PGS from the same host (rabbit IgG) for double staining in combination with different fluorescently tagged anti rabbit antibodies. This can be tricky and is prone to staining artifacts. E.g, what is the explanation for the ACE2 positive staining within the tubule structures? Why does the virus stain appear on the outer edge relative to receptor? These patterns should be confirmed with a distinct combination of antibodies. Along similar lines, the staining for DP2 and PDS appear very dim and different between patient samples making results inconclusive.

More minor issues:

1. Table 1 lists 5 patients that perished; what is the 6th autopsy case? It is also listed that 3 AKI cases and 2 non AKI cases perished, and that 2 AKI patients and 3 non AKI cases had chronic renal disorders. It is unclear how these relate to the autopsy samples examined.
2. The scale bar in Fig. 1 columns is different in the HBV-MN biopsy and trauma autopsy tissue relative to COVID19 tissue.
3. There are still numerous English language issues throughout; the term lymphocyte is used in numerous occasions, where macrophages are included in infiltrates.

Reviewer #4 (Remarks to the Author):

The authors addressed nicely the major concerns as well as the minor points. However some points still need to be clarified.

Q1: Immunofluorescent staining showed that the expression of hypoxic damage molecules including DP2 and PDS was induced in COVID19-kidney tubules. Was it also observed within the histopathological controls?

Q3: The authors replied that they do not have a P3 level virology lab in order to assess COVID19 tissue viral load by PCR per reference gene. So, they have to collaborate with a virology lab that is able to assess it.

Q8: It is stated "We did not have control postmortems to do TEM observation". Sure, but it can be done easily by including 2 new postmortem controls in order to assess TEM findings in non-COVID19 deceased patients.

Point to point response to reviewers' comments

REVIEWER COMMENTS

Reviewer #3 (Remarks to the Author):

This revised manuscript addressed some concerns raised by the previous review, including more extensive data on the patient cohort, preexisting conditions, treatment, etc. The main novelty continues to reside in evaluation of kidney autopsy material from 6 post mortem cases linking viral antigen expression with tubule damage. New data is provided with respect to COVID19 and control tissues, as well as expression of viral antigen in relation to ACE2 and DP2/PDS to provide a correlation between location of viral antigen and tissue damage. However, there are ongoing and new concerns with respect to nucleocapsid antigen staining, costaining, and interpretation with respect to leukocyte infiltrates and complement deposition, leaving conclusions tentative.

Major ongoing concerns:

Q1. Time between autopsy and death should be included in Methods as long interval may affect results.

A1: We have added the associated information to the Methods: “The perished patients were taken postmortem examination immediately (4 cases) or kept in -20°C till postmortem within 24 hours after death (2 cases).”

Q2. Fig. 2 only shows EM for the same section of one patient sample (not 2 patient samples as indicated) and the 500 nm Figures appear the same. It is still unclear what structural parameters were chosen for the arrow highlighted densities termed as virus like.

A2: In the revised manuscript, the TEM of two cases (#2 and #4) were added, and Coronavirus-like particles were identified in the cytoplasm of renal proximal tubular epithelium. The diameter of virus-like particles varied from about 60 nm to 120 nm, with distinctive spikes, around 20 to 25 nm, presenting in a solar “corona” appearance. Additional features of this coronavirus included adjacent double membrane with

surface projections, nucleocapsid apposing to the viral envelope, and the interior electron-lucent of the particles. In addition, sections from trauma victim were also used as negative control and no coronavirus-like particles were observed. Hopefully these modifications are adequate.

Q3. The nucleocapsid stain in Fig. 3 remains fuzzy and not very specific despite the negative controls. The interpretation of a viral inclusion body also remains tentative as it appears too large. Have the authors evaluated other anti N or S antibodies to confirm staining patterns?

A3: We ordered two novel mouse monoclonal antibodies, the anti-SARS NP antibodies (ab273434, 1:500, mouse monoclonal 6H3, Abcam) and anti-SARS spike glycoprotein (S) antibodies (ab273433, 1:500, mouse monoclonal 1A9, Abcam) , which can react with SARS-CoV-2 viral NP and S antigens, respectively, to detect SARS-CoV-2 in human kidney by immunohistochemistry. Our results demonstrated that both SARS-CoV-2 viral NP and S antigens were seen in the kidney from COVID-19 autopsies (**Fig. 3b**). Thank you very much.

Q4. Figures 4 and 5 use primary anti N, ACE2, DP2 and PGS from the same host (rabbit IgG) for double staining in combination with different fluorescently tagged anti rabbit antibodies. This can be tricky and is prone to staining artifacts. E.g, what is the explanation for the ACE2 positive staining within the tubule structures? Why does the virus stain appear on the outer edge relative to receptor? These patterns should be confirmed with a distinct combination of antibodies. Along similar lines, the staining for DP2 and PDS appear very dim and different between patient samples making results inconclusive.

A4: In the revised manuscript, antibodies used for immunofluorescent double staining are from different hosts. For examples, the anti-SARS-CoV-2 NP antibodies (ab273434, 1:500, mouse monoclonal 6H3, Abcam) and anti-ACE2 (clone ID: 10108-RP01, 1:100, rabbit IgG; Sino Biological) or anti-SARS S antibodies (ab273433, 1:500, mouse monoclonal 1A9, Abcam) and anti-ACE2 (clone ID: 10108-RP01, 1:100, rabbit IgG), respectively, were used to detected the co-expression

of SARS-CoV-2 NP and S antigen and ACE2 in human kidney tubules (**Fig. 4b**). From our immunohistochemistry and immunofluorescent staining, the expression of ACE2 restricts on the brush border of proximal tubular cells (**Fig. 4a**), whereas SARS-CoV-2 antigens accumulate in cytoplasm (**Fig. 3b**), so the SARS-CoV-2 viral antigen staining appear on the outer edge relative to ACE2 receptor (**Fig. 4b**).

The status of SARS-CoV-2 antigen and PD2 and PDS proteins in kidney tubules were analyzed by immunofluorescent double staining with using novel antibodies. Our results illustrated that SARS-CoV-2 can augment the expression of PD2 and PDS in infected kidney tubules, and the staining for DP2 and PDS is better than previous (**Fig. 5a and 5b**). Hopefully these revisions are accepted.

More minor issues:

Q1. Table 1 lists 5 patients that perished; what is the 6th autopsy case? It is also listed that 3 AKI cases and 2 non AKI cases perished, and that 2 AKI patients and 3 non AKI cases had chronic renal disorders. It is unclear how these relate to the autopsy samples examined.

A1: The six cases of COVID-19 autopsies are from Jinyintan Hospital in Wuhan city, whereas data in **Table 1** are from patients in General Hospital of Central Theatre Command in Wuhan. In the early time of epidemic, only the perished patients in Jinyintan Hospital, which is a special hospital for infectious diseases, are permitted to perform postmortem by Chinese government. We have also indicated these messages in the main manuscript, please check it in the Method section. Thank you very much!

Q2. The scale bar in Fig. 1 columns is different in the HBV-MN biopsy and trauma autopsy tissue relative to COVID19 tissue.

A2: The figures from HBV-MN biopsy and trauma autopsy were re-taken by the same microscope, now the scale bar in all of these figures are similar now. Thank you very much!

Q3. There are still numerous English language issues throughout; the term lymphocyte is used in numerous occasions, where macrophages are included in infiltrates.

A3: Thanks for the comment and we have tried our best to revise this manuscript, not only invited a native English speaker to revise our manuscript, but also submitted the manuscript to American Journal Expert to edit the English expression. We also pay our attention to lymphocyte and macrophages. We believe these modifications will improve the quality of this manuscript and make it much clearer in its presentation to readers. Compared with the original manuscript, all changes in the revised manuscript have been clearly indicated in red fonts. Hopefully these modifications are acceptable.

Point to point response to reviewers' comments

REVIEWER COMMENTS

Reviewer #4 (Remarks to the Author):

The authors addressed nicely the major concerns as well as the minor points. However some points still need to be clarified.

Q1: Immunofluorescent staining showed that the expression of hypoxic damage molecules including DP2 and PDS was induced in COVID19-kidney tubules. Was it also observed within the histopathological controls?

A1: In the revised manuscript, we re-did the experiments and immunofluorescent staining showed that the expression of DP2 and PDS in COVID19-kidney tubules was enhanced by SARS-CoV-2 infection, whereas, both DP2 and PDS expressions are absence in kidney tissues from HBV-MN. Hopefully these revisions are acceptable.

Q2: The authors replied that they do not have a P3 level virology lab in order to assess COVID19 tissue viral load by PCR per reference gene. So, they have to collaborate with a virology lab that is able to assess it.

A2: In the revised manuscript, *In situ* RNAscope was used to detect SARS-CoV-2 viral RNA and results clearly demonstrated that SARS-CoV-2 in kidney tubules from COVID-19 autopsies (**Fig. 3a**). TEM observation (**Fig. 2**) and both SARS-CoV-2 NP and S antigen were observed in kidney tubules by immunohistochemistry (**Fig. 3b**), our data demonstrate that SARS-CoV-2 can infect human kidney tubules. Thank you very much.

Q3: It is stated "We did not have control postmortems to do TEM observation". Sure, but it can be done easily by including 2 new postmortem controls in order to assess TEM findings in non-COVID19 deceased patients.

A3: In the revised manuscript, the TEM of two cases (#2 and #4) were added, and Coronavirus-like particles were identified in the cytoplasm of renal proximal tubular

epithelium. The diameter of virus-like particles varied from about 60 nm to 120 nm, with distinctive spikes, around 20 to 25 nm, presenting in a solar “corona” appearance. Additional features of this coronavirus included adjacent double membrane with surface projections, nucleocapsid apposing to the viral envelope, and the interior electron-lucent of the particles. In addition, sections from HBV-MN were also used as negative control and no coronavirus-like particles were observed. Hopefully these modifications are adequate. Hopefully these modifications are adequate.

REVIEWERS' COMMENTS

Reviewer #3 (Remarks to the Author):

This 2nd revision addresses all concerns of the second review by including additional control experiments requested and addressing other concerns. The authors carefully evaluated other anti N and S, as well as novel DP2 and PDS antibodies to confirm staining patterns. The only new data that appear somewhat surprising are the strong in situ RNA scope signals obtained for the S-sense probe, which appear stronger than the S probes (Fig 3a). As the minus strand mRNAs are thought to be at lower abundance than the positive ones during CoV replication, this may be worth a further comment in line 168.

Minor corrections:

1. Fig4b: Line 186 states that sections from control are 'absent' for RS CoV-2 NP and S. This statement should be dampened. Although the COVID-19 samples exhibit a much more pronounced and robust staining pattern, some punctate red NP staining is evident in controls, which may reflect some nonspecific staining.
2. line 130 'leukocyte' not lymphocyte

Reviewer #4 (Remarks to the Author):

In this revised version of their manuscript the authors nicely addressed my questions. No further comments.

Point to point response to reviewers' comment

Reviewer #3 (Remarks to the Author):

This 2nd revision addresses all concerns of the second review by including additional control experiments requested and addressing other concerns. The authors carefully evaluated other anti N and S, as well as novel DP2 and PDS antibodies to confirm staining patterns. The only new data that appear somewhat surprising are the strong in situ RNA scope signals obtained for the S-sense probe, which appear stronger than the S probes (Fig 3a). As the minus strand mRNAs are thought to be at lower abundance than the positive ones during CoV replication, this may be worth a further comment in line 168.

A: Thanks for your kind comments. As it can be seen in Fig3a, there are two different sections where the probe-S and the probe-S-sense is staining. So we think that only two pictures is not sufficient to draw certain conclusion. Actually, the positive staining of probe-S-sense is comparable or a little bit stronger than probe-S in the whole experiment data, please kindly check the Table 2.

Minor corrections:

1. Fig4b: Line 186 states that sections from control are 'absent 'for RS CoV-2 NP and S. This statement should be dampened. Although the COVID-19 samples exhibit a much more pronounced and robust staining pattern, some punctate red NP staining is evident in controls, which may reflect some nonspecific staining.

A: Thanks for your carefulness. The corresponding statement has been revised, please check it. For your convenience, I have attached it here. "The sections from HBV-MN and trauma victims' biopsies are also positive for ACE2, however, these sections are rarely stained by anti-NP or anti-S antibodies, especially in the

kidney tubules (Fig. 4b).”

2. line 130 'leukocyte' not lymphocyte

A: Thanks for your carefulness. The corresponding statement has been revised.

Reviewer #4 (Remarks to the Author):

In this revised version of their manuscript the authors nicely addressed my questions. No further comments.

A: Thanks for your kind approval.

Point to point response to reviewers' comments

We are sorry to contact you again asking for the inclusion of additional data after offering to accept your manuscript in principle. However, serious issues have been raised by one of the original peer reviewers after our previous decision. We are aware that these comments should have been raised during the peer review process and this is an extremely unfortunate and rare event. After discussing with another reviewer, we came to the conclusion that two main issues are left unsolved:

Q1. In situ hybridization data. The in situ hybridization reactions using a SARS-COV-2-specific probe might cause false positive results. Therefore, please provide a non-COVID-19 control for the sense and antisense SARS-COV-2 probes used in Fig.3a. Control non-COVID19 kidneys must be shown in a side-by-side fashion, as performed for the other experiments. Please note that using probes against other transcripts does not represent the the best negative control.

A1: Thanks for reviewers' comments and we have tried our best to revise this manuscript. Taking the considerations of the reviewer, we repeated RNAscope (*in situ* hybridization) multiple times with inclusion of the kidney sections from HBV-MN patients as the negative controls. Our results clearly demonstrated that kidneys from COVID-19 autopsies are positive for SARS-CoV-2 RNA (Fig. 3a), whereas, the virus RNA is undetectable in the sections from HBV-MN patients (Supplementary Figure 2). Hopefully these modifications are adequate.

2. Electron Microscopy data. Very recent articles highlighted the difficulties in directly distinguishing CoV particles from other cellular vesicles/organelles. (including, but not limited to <https://onlinelibrary.wiley.com/doi/10.1002/path.5547>, [https://www.thelancet.com/journals/lancet/article/PIIS0140-6736\(20\)311880/fulltext](https://www.thelancet.com/journals/lancet/article/PIIS0140-6736(20)311880/fulltext) <https://jasn.asnjournals.org/content/31/9/2223>). It does appear that the higher resolution 'viral' structure in cytoplasm according to the authors interpretation (line 153) is not virus but rather resembles the structures highlighted in the J Pathol review article Fig. 4. If in the cytoplasm the virus should not have nicely formed 'corona', as virus assembles in membrane bound structures. Another issue may be the 'electron lucent' core (line 157 of the manuscript) in Fig 2. Viral cores should have more punctate electron dense structures characterizing viral nucleocapsid protein. The best way to assess viral particles would be to use immunogold tagged Ab.

A2: Thanks for these helpful comments. We actually have conducted transmission electron microscopy (TEM) experiments for several times using different TEM resource labs during the past three months. Successfully, we are able to present novel TEM data in the revised manuscript.

In Fig. 2, we showed that virus-like particles are surrounded by double membrane structures, which located around Golgi apparatus (G) (Fig. 2a, b). This is in agreement with the organelle location of the viral replication. The diameter of the particles is around 60-80 nm, and some granular electron-dense interiors are also observed (red arrows in Fig. 2a~c), which are believed to be nucleocapsid protein. The surface projections (spikes) are not apparent but sometimes a “fuzz” is seen around some particles. Meanwhile, we would like to bring up in fact that our TEM results are consistent with other recent reports in the literatures (Martines RB,et.al. *Emerg Infect Dis.* 2020 Sep;26(9):2005-2015; Yao XH, et.al. *Cell Res.* 2020 Jun;30(6):541-543; Meinhardt J, et al. *Nat Neurosci.* 2020;10.1038/s41593-020-00758-5). Collectively, these TEM observations, together with other sets of supporting data, suggest these particles are likely to be the intracellular viruses.

The virus like particles that we have observed in the infected human kidney tissues are dissimilar to the following particular non-virus particle structures that have been reported in the literature (Neil D, et.al. *J Pathol.* 2020 Dec;252(4):346-357; Meinhardt J, et al. *Nat Neurosci.* 2020;10.1038/s41593-020-00758-5):

(1) It is noteworthy that under TEM, these virus-like particles in the infected kidney appear to locate near the Golgi apparatus rather than nucleus or cytomembrane, making them different from nuclear pores or microvilli; (2)The interiors of the observed virus-like particles are not hollow as vesicles, making them different from multi-vesicular bodies (MVB) or coated vesicles; (3) The intraluminal vesicles of MVB are usually with a clathrin coat, giving a spikey appearance, and this is not the case of virus-like particles observed in our study; (4) The virus-like particles of our study showed relative uniformity in size, making them different from endosomal structures or other cell trafficking micro-vesicles; (5) Rough endoplasmic reticulum cisternae may also mimic coronavirus in some cross-sectional orientations , however, they all have granular appearance of ribosomes both on the outer or inner membrane and the particles are range in size, this is not the case of our virus-like particles neither.

It is worthy to note that the human tissues (especially autopsy specimens) are different from cultured cells. In perished cases, ARDS and viral sepsis are relatively common, many cells are necrotic and disintegrated. Moreover, the ultrastructure features of the virus in vivo are not as conspicuous as that in cultured cells, especially the nucleocapsid protein.

We agree with the reviewer that positive data from immuno-EM will be more definitive in validating virus components that colocalize with the observed virus like particles. In fact, two of the best TEM resource labs in China have helped us to perform immuno-EM experiments.

Unfortunately, the results are not satisfactory, one possibility is the weakening of antigenicity after fixation by formalin and long-term preservation in paraffin wax at room temperature, as well as the destruction of protein structure by ultrathin slicing, it is very difficult to get positive results, even the lung tissues from post-mortem. Anyhow we don't think these negative data are sufficient to refute our conclusion which is supported with multiple angled testing results.

REVIEWERS' COMMENTS

Reviewer #3 (Remarks to the Author):

The two major concerns have been adequately addressed.

1. New Figures are presented for TEM; the distinction to cellular organelles are pointed out and discussed, and efforts to perform immuno EM included in the discussion.
2. the RNA in situ controls are also provided

Reviewer #5 (Remarks to the Author):

In the rebuttal there are listed 3 articles that illustrate that the "TEM results are consistent with other recent reports in the literatures." However, the article by Yao, et al. does not show EM images of coronavirus but rather images of coated vesicles that have been misidentified as virus. The articles by Martines, et al., and Meinhardt, et al., are both good and show coronaviruses. A third example would be an article by C. Dittmayer, et al., Lancet. 396(10260):e64-e65, 2020 10 31, titled "Why misinterpretation of electron micrographs in SARS-CoV-2-infected tissue goes viral."

For the image in figure 2c, although you have argued against it, I believe that these are nuclear pores. Since there is not a nucleus visible close-by, it could be that the plane of section just skimmed the nuclear envelope. Regardless, these could not be coronavirus particles because they are free in the cytoplasm and are not enclosed within a membrane-bound compartment. The structures in figure 2d are coated vesicles and, again, they are free in the cytoplasm and are not within a membrane-bound compartment.

I struggled with the virus-like particles in figures 2a and 2b because they are spherical and are within a membrane-bound compartment. And, admittedly, they look quite similar to the virus particles shown in the article by Meinhardt, et al. However, the tissue in this paper had been excised from a formalin fixed, paraffin-embedded block, deparaffinized, rehydrated and processed to epoxy resin. This technique can greatly compromise the morphology of the tissue, and one reason these particles can be diagnosed as coronaviruses is due to the large number of virions. In the present manuscript the virus-like particles are limited in number, and also do not show the dot-like structures of the ribonucleoprotein, so that they are not convincing as being coronavirus particles. The morphology of the virus particles in the other two articles, particularly the one by Dittmayer, et al., are better examples of what should be expected and I recommend that the authors look closely at this paper.

Point to point response to reviewers' comments

Reviewer #3 (Remarks to the Author):

The two major concerns have been adequately addressed.

1. New Figures are presented for TEM; the distinction to cellular organelles are pointed out and discussed, and efforts to perform immuno EM included in the discussion.
2. the RNA in situ controls are also provided

A : Thanks for the comments.

Reviewer #5 (Remarks to the Author):

In the rebuttal there are listed 3 articles that illustrate that the “TEM results are consistent with other recent reports in the literatures.” However, the article by Yao, et al. does not show EM images of coronavirus but rather images of coated vesicles that have been misidentified as virus. The articles by Martines, et al., and Meinhardt, et al., are both good and show coronaviruses. A third example would be an article by C. Dittmayer, et al., Lancet. 396(10260):e64-e65, 2020 10 31, titled “Why misinterpretation of electron micrographs in SARS-CoV-2-infected tissue goes viral.”

For the image in figure 2c, although you have argued against it, I believe that these are nuclear pores. Since there is not a nucleus visible close-by, it could be that the plane of section just skimmed the nuclear envelope. Regardless, these could not be coronavirus particles because they are free in the cytoplasm and are not enclosed within a membrane-bound compartment. The structures in figure 2d are coated vesicles and, again, they are free in the cytoplasm and are not within a membrane-bound compartment.

I struggled with the virus-like particles in figures 2a and 2b because they are spherical and are within a membrane-bound compartment. And, admittedly, they look quite similar to the virus particles shown in the article by Meinhardt, et al. However, the tissue in this paper had been excised from a formalin fixed, paraffin-embedded block, deparaffinized, rehydrated and processed to epoxy resin. This technique can greatly compromise the morphology of the tissue, and one reason these particles can be diagnosed as

coronaviruses is due to the large number of virions. In the present manuscript the virus-like particles are limited in number, and also do not show the dot-like structures of the ribonucleoprotein, so that they are not convincing as being coronavirus particles. The morphology of the virus particles in the other two articles, particularly the one by Dittmayer, et al., are better examples of what should be expected and I recommend that the authors look closely at this paper.

A: We totally agree with your professional opinion. To ensure accuracy of this manuscript, we removed the electron microscopy data, and discussed in the discussion section, which is in line with the editor's request. Thank you.